# Mechanism of beta-arrestin 1 mediated Src activation via Src SH3 domain revealed by cryo-electron microscopy

Natalia Pakharukova[1,2], Brittany N. Thomas[1,2], Harsh Bansia[3,4], Linus Li[1], Dana K. Bassford[1,2], Rinat R. Abzalimov[5], Jihee Kim[1], Alem W. Kahsai[1], Biswaranjan Pani[1], Kunhong Xiao[6,7], Roni Ochakovski[1], Shibo Liu[5], Xingdong Zhang[1], Seungkirl Ahn[1], Amedee des Georges[3,4,5] ✉ & Robert J. Lefkowitz[1,2,8] ✉

Beta-arrestins (βarrs) are key regulators and transducers of G-protein coupled receptor signaling; however, little is known of how βarrs communicate with their downstream effectors. Here, we delineate structural mechanisms underlying βarr-mediated signal transduction. Using cryo-electron microscopy, we elucidate how βarr1 recruits and activates the non-receptor tyrosine kinase Src, a well-established signaling partner of βarrs. βarr1 engages Src SH3 through two distinct sites, each employing a different recognition mechanism: a polyproline motif in the N-domain and a non-proline-based interaction in the central crest region. At both sites βarr1 interacts with the aromatic surface of SH3, disrupting the autoinhibited conformation of Src and directly triggering its allosteric activation. This structural evidence establishes βarr1 as an active regulatory protein rather than a passive scaffold and suggests a potentially general mechanism for βarr-mediated signaling across diverse effectors.

Beta-arrestins 1 and 2 (βarr1 and βarr2) are ubiquitously expressed scaffold proteins that interact with most, if not all, G-protein coupled receptors (GPCRs)[1]. βarrs mediate receptor desensitization and intracellular trafficking, as well as initiate downstream signaling cascades, independently or in concert with G-proteins[2]. In their role as signal transducers, βarrs link GPCRs with numerous downstream effectors, including components of several mitogen-activated protein kinase cascades and Src family kinases, amongst many others[2].

Early studies proposed that βarrs function primarily as passive signaling scaffolds; however, more recent work has demonstrated that βarrs can directly and allosterically activate specific effector enzymes, including the non-receptor tyrosine kinase Src[3,4], the mitogen-

activated protein kinases c-Raf[5] and extracellular signal-regulated kinase 2[6]. Despite these advances, the molecular mechanisms underlying βarr-mediated signaling remain poorly understood, largely due to the absence of high-resolution structural information for complexes formed between βarrs and their diverse effector enzymes. Structural characterization of these interactions has been limited by their transient nature and mid-micromolar binding affinities, necessitating tailored strategies for complex stabilization.

Src homology 3 (SH3) domains are one of the most ubiquitous binding modules with nearly 300 members found in the human genome[7]. SH3-containing proteins are involved in various signaling pathways, such as cell growth and proliferation, endocytosis, and

[1]Department of Medicine, Duke University Medical Center, Durham, NC 27710, USA. [2]Howard Hughes Medical Institute, Duke University Medical Center, Durham, NC 27710, USA. [3]Department of Molecular Pathobiology, College of Dentistry, New York University, New York 10010, USA. [4]Pain Research Center, New York University, New York 10010, USA. [5]Structural Biology Initiative, CUNY Advanced Science Research Center, New York, NY 10031, USA. [6]Center for Proteomics & Artificial Intelligence, Allegheny Health Network Cancer Institute, Pittsburgh, PA 15202, USA. [7]Department of Biomedical Engineering, College of Engineering, Carnegie Mellon University, Pittsburgh, PA 15213, USA. [8]Department of Biochemistry, Duke University Medical Center, Durham, NC 27710, USA. ✉e-mail: a.des.georges@nyu.edu; lefko001@receptor-biol.duke.edu

cytoskeleton remodeling[8]. Notably, βarrs have been shown to interact with the SH3 domains of several proteins[3], including the proto-oncogene kinase Src[9]. Intriguingly, βarrs lack the canonical left-handed type II polyproline motifs required for SH3 binding; therefore, how βarrs recruit SH3 remains unknown.

Here, we report the structural insights into the fundamental mechanisms underlying βarr-mediated signal transduction. Using Src as a prototypical effector of βarr, we elucidate the molecular basis for Src SH3 recruitment and Src allosteric activation by cryo-electron microscopy (cryo-EM).

## Results

### βarr1 uses two distinct sites to bind SH3

To map the binding interface between the SH3 domain of Src and βarr1, we utilized a disulfide trapping strategy[10]. The βarr1 and SH3 Src sequences were derived from rat (Rattus norvegicus) and chicken (Gallus gallus), respectively. Both proteins exhibit high inter-species sequence similarity−93.54% for βarr1 and 93.66% for Src (100% for Src

SH3 domain), with the most notable sequence variations present in the unique domain of Src and at the C-terminus of βarr1 (Supplementary Fig. 1). We introduced cysteine substitutions at 18 different positions in βarr1 in proximity to P88-P91 and P120-P124, previously reported to be critical for βarr1−Src interaction (Fig. 1a)[3,9]. Guided by the available structures of SH3 domains in complex with polyproline-rich peptides, we designed 13 cysteine mutants of SH3 (Fig. 1b). We comprehensively tested 234 combinations of purified βarr1 and SH3 cysteine mutant pairs by inducing disulfide bond formation with hydrogen peroxide in vitro. Active βarr1 was shown to bind more strongly to SH3[4]; therefore, prior to disulfide trapping reactions βarr1 was activated using a synthetic phosphopeptide mimicking the C-tail of vasopressin 2 receptor (V2Rpp) and the stabilizing antibody fragment Fab30. Intriguingly, we observed the formation of disulfide cross-linked SH3−βarr1−V2Rpp−Fab30 complexes by two βarr1 mutants, E92C in the distal part of the N-domain (hereafter referred to as βarr1-N site) and P120C in the central crest region (hereafter referred to as βarr1-CC site) (Fig. 1c, d). In contrast, little to no SH3−βarr1 cross-linked

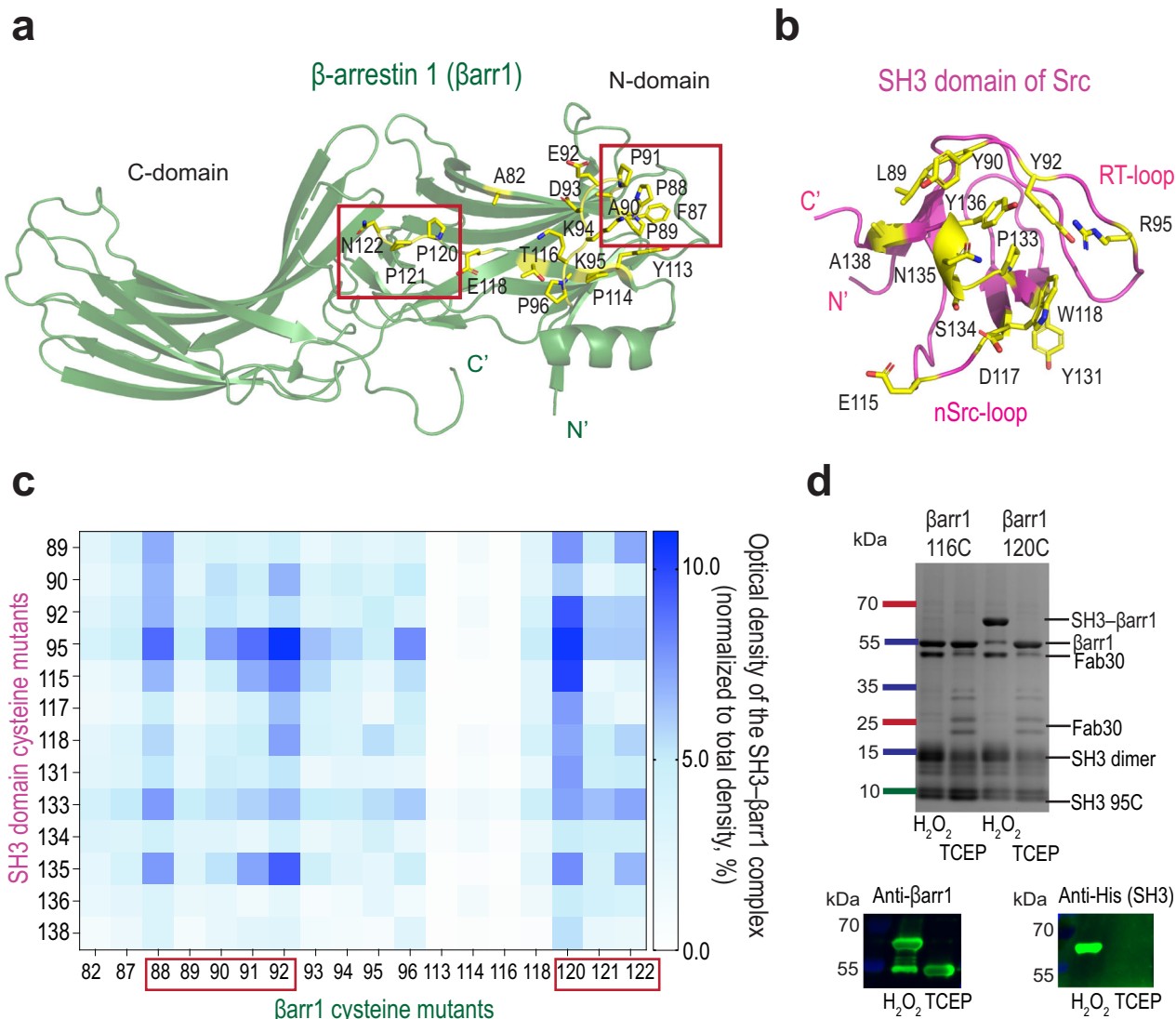

**Fig. 1 | β-arrestin 1 (βarr1) uses two distinct sites to bind SH3. a, b** Structures of βarr1 (green, PDB: 4JQI) and SH3 (magenta, PDB: 2PTK), cartoon representation. Residues used for disulfide trapping are shown as yellow sticks and labeled. Proline regions of βarr1 are framed in rectangles. **c** Heatmap of different SH3−βarr1 cysteine-pair complexes; densitometry analysis of Coomassie blue gels. The SH3−βarr1 complex band was normalized to the total density. Mean values from $n = 3$ independent

experiments are shown; the gels were processed in parallel. **d** Top: representative results of disulfide trapping with βarr1_120C (binding site for SH3) and βarr1_116C (non-binding site for SH3) (Coomassie blue gel). Bottom: Western blot of disulfide trapping reaction between βarr1_120C and SH3_R95C confirms that the 60-kDa band is the SH3−βarr1 complex. Data are representative of $n = 3$ independent experiments (TCEP - Tris-(2-Carboxyethyl)phosphine, Hydrochloride).

complexes were formed by other βarr1 mutants, such as T116C (Fig. 1d). The SH3 mutant R95C in the RT-loop gave the strongest disulfide cross-linked band in both sites. As βarr1-N and βarr1-CC sites are more than 20 Å apart, we hypothesized that the SH3 domain might independently bind at two sites of βarr1. Complex formation at the βarr1-CC site was more efficient in the presence of V2Rpp and Fab30, whereas the βarr1-N site forms the complex equally well regardless of activation (Supplementary Fig. 2a).

SH3 E115C also formed a cross-linked complex with βarr1-CC (Fig. 1c). Analytical size-exclusion chromatography revealed that the SH3_R95C–βarr1_P120C complex displays a more prominent and symmetric peak compared to SH3_E115C–βarr1_P120C, suggesting greater homogeneity and stability (Supplementary Fig. 2b). βarr1 mutants P88C and P91C mutants also formed the cross-linked complex with SH3_R95C, but to a lesser degree than E92C suggesting the dynamic nature of the SH3–βarr1 interfaces (Fig. 1c).

To validate whether V2Rpp-activated βarr1 can bind SH3 at two distinct sites, we used isothermal titration calorimetry (Supplementary Fig. 2c). The integrated heat curve revealed two binding events: one with a Kd of ~6 μM and the other with a significantly lower affinity of ~50 μM. The favorable entropy (-TΔS) suggests that in both sites, SH3–hydrophobic interactions drive βarr1 binding. Furthermore, the favorable enthalpy (ΔH) at the higher affinity site indicates some contribution from hydrogen bonding and van der Waals forces.

## βarr1 binds to the aromatic surface of SH3

To elucidate the mechanism of SH3 recruitment by βarr1, we formed cross-linked complexes with the best mutant pairs SH3_95C–βarr1_120C (SH3–βarr1-CC) and SH3_95C–βarr1_92C (SH3–βarr1-N) and determined the structures by cryo-EM (Fig. 2a–c, Supplementary Fig. 3, Supplementary Fig. 4, Supplementary Table 1). The map obtained from the SH3–βarr1-CC complex has a global resolution of 3.47 Å and showed a rigid interaction with SH3 in the central crest area of βarr1 (Fig. 2a, Supplementary Fig. 4e). The complex was activated by V2Rpp and stabilized by Fab30 and nanobody 32 (Nb32)[11]. βarr1 adopts an active conformation[12], characterized by C-tail displacement and inter-domain rotation, and binds to the aromatic surface of SH3 containing residues from the RT loop, nSrc loop, and the $3_{10}$ helix (Fig. 2b, Supplementary Video 1). βarr1 interacts with SH3 using β-strand V (residues 75-80) in the N domain and a part of the lariat loop in the C domain. Y90 and Y136 in SH3 interact with F75 in βarr1. The side chains of Y136 and N135 in SH3 are well-placed to make hydrogen bonds with the main chain oxygen of R76 in βarr1. Additional hydrogen bonds are possible between the main chain oxygen of E93 in SH3 and the side chain of N122 in βarr1, as well as between W118 in SH3 and the main chain oxygen of D78 in βarr1.

The SH3–βarr1-N complex was formed with V2Rpp and Fab30 but without Nb32, as Nb32 hinders complex formation due to the steric clashes with SH3. The cryo-EM map showed SH3 bound to the distal part of the N-terminal domain of βarr1 (Fig. 2c). Whereas SH3–βarr1-CC has a clearly resolved density for SH3, in SH3–βarr1-N the density for SH3 is weaker (local resolution is 4.2–4.6 Å) since SH3 binds to a flexible loop of βarr1, the [87]FPPAPEDK[94] motif (Fig. 2c, Supplementary Fig. 4f). We used molecular dynamics flexible fitting (MDFF) constrained by the position of the disulfide bond between βarr1 and SH3 to place the main chain of the SH3 domain. The proline region of the βarr1-N site (P88, P89, P91) appears to interact with the aromatic surface of SH3, similarly to the βarr1-CC site (Fig. 2c, Supplementary Video 2). However, the weak density for SH3 in the SH3–βarr1-N complex prevents confident identification of the specific SH3 residues involved in βarr1 binding. To confirm the interaction interfaces, we introduced mutations in SH3 and βarr1 and tested the binding by pull-down assay. Mutations in the RT loop (Y90A, Y92C) and the $3_{10}$ helix (Y136A) of SH3 drastically reduced its ability to bind βarr1 (Fig. 2d). Likewise, the substitutions of prolines in βarr1 impair SH3 binding

(Fig. 2e). These findings, together with the disulfide trapping data indicating that the RT loop faces the SH3–βarr1 interaction interface, confirm that residues within both the RT loop and $3_{10}$ helix are critical for the SH3–βarr1 interaction at both sites of βarr1.

The structural and disulfide trapping data indicate a highly dynamic mode of SH3 binding to the βarr1-N site (Fig. 1c), with disulfide trapping capturing only a single state of this inherently dynamic interaction. Cross-linked complex formation by several SH3–βarr1 cysteine pairs (Fig. 1c) suggests that SH3 might adopt alternative orientations relative to the βarr1-N site, highlighting the complexity of low-affinity interactions in signaling cascades.

Notably, βarr1 uses a distinct recognition mechanism for SH3 binding at each site. The βarr1-N site binds SH3 exclusively via proline residues. In contrast, the recruitment of SH3 via the βarr1-CC site is driven by non-proline residues in β-strand V and the lariat loop, as well as two proline residues after β-strand VI.

Since the SH3–βarr1 complexes were stabilized using cysteine substitutions and disulfide trapping, we sought to determine whether these interactions also occur under physiological conditions, in the absence of engineered mutations. To address this, we mapped the interaction interfaces and dynamics of free βarr1, V2Rpp-activated βarr1, and SH3 using hydrogen–deuterium exchange mass spectrometry (HDX-MS) (Fig. 3; Supplementary Table 2; Supplementary Data 1–6). HDX-MS provides insights into protein interaction interfaces by measuring deuterium uptake in peptides typically 5–15 residues in length. Thus, this approach enables us to verify whether SH3 binds to two distinct regions of βarr1 in vitro without disulfide trapping. Approximately 5% of the analyzed peptides displayed a statistically significant decrease in deuterium uptake in the presence of SH3, defined as a statistically significant change greater than 0.3 Da for free βarr1 and 0.36 Da for βarr1–V2rpp (Supplementary Tables 2–3, Supplementary Data 1–2, 5). Consistent with the structures and the pull-down data, βarr1 showed a significant decrease in HDX rate in both the βarr1-CC and βarr1-N sites in the presence of SH3 (Fig. 3a, b; Supplementary Table 3). Although the differences in deuterium uptake for peptides corresponding to the βarr1-CC and βarr1-N sites were modest (0.2–0.8 Da), consistent with low-affinity interactions and partial complex occupancy arising from dilution during HDX labeling, the changes were reproducible across multiple time points and supported by several overlapping peptides in both free and V2Rpp-activated βarr1 (Supplementary Table 3; Fig. 3a, b; Supplementary Fig. 5a). Within these regions, longer peptides spanning both βarr1-CC and -N sites showed more pronounced changes, while shorter internal peptides displayed attenuated effects due to key interacting residue being positioned at the peptide N-terminus and increased back exchange (Supplementary Fig. 5a). Notably, a decrease in deuterium uptake was also observed for residues 34–47, which do not directly interact with SH3 in the structure but are located near the βarr1-N site, suggesting that SH3 binding may indirectly stabilize this region. Interestingly, reduced deuterium uptake was also detected in the hinge region of βarr1, in the C-domain spanning residues 257–310, and within the C-terminal tail (352–377) in V2Rpp-activated βarr1 (Fig. 3a). Since Src does not directly interact with the C-domain of βarr1[9], the reduced deuterium uptake in this region likely arises from long-range allosteric stabilization propagated through SH3 binding.

We next performed a qualitative HDX-MS experiment to assess changes in the deuterium uptake of SH3 in the presence of either free or V2Rpp-activated βarr1 (Supplementary Tables 2 and 4; Supplementary Fig. 5b–d; Supplementary Data 3–4, 6). Consistent with the structural data, peptides encompassing the nSrc loop and the $3_{10}$ helix of SH3 exhibited decreased deuterium uptake, with more pronounced differences observed in the presence of V2Rpp-activated βarr1 (Supplementary Table 4; Supplementary Fig. 5b, c). In contrast, peptides corresponding to the RT loop predominantly showed increased deuterium uptake, likely reflecting local destabilization and enhanced

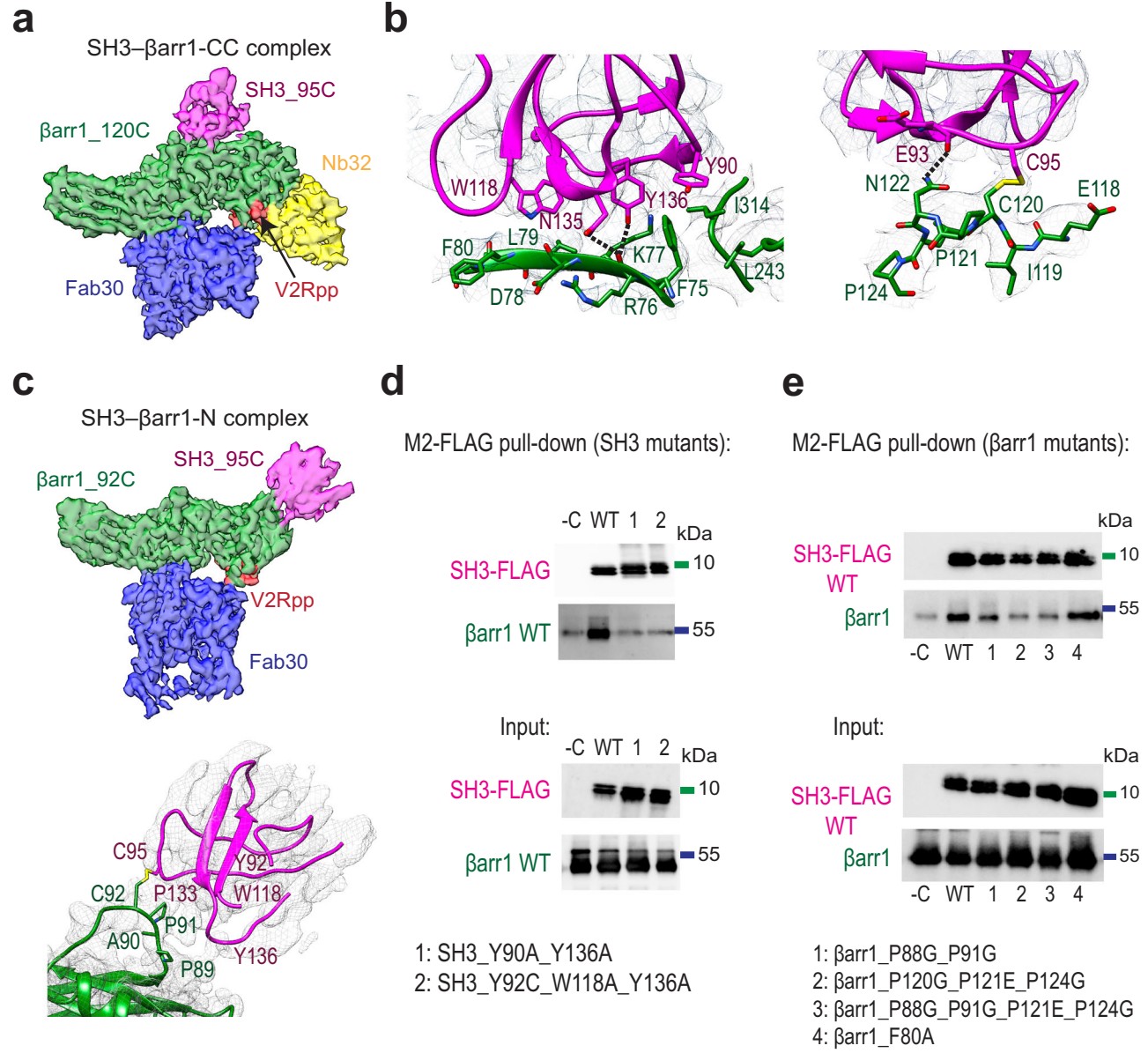

**Fig. 2 | βarr1 binds the aromatic surface of SH3. a** Cryo-EM density map of the SH3–βarr1-CC complex (green, βarr1; magenta, SH3; red, phosphopeptide mimicking the C-tail of vasopressin 2 receptor (V2Rpp); blue, Fab30; yellow, nanobody 32 (Nb32)). Map contour level is 0.75. **b** Interaction interface of SH3–βarr1-CC complex (green, βarr1; magenta, SH3) with density mesh (upsample map 0.69 Å/pix, contour level 0.37). Hydrogen bonds are shown as dashed lines. See also Supplementary Video 1. **c** Top panel: Cryo-EM density map of the SH3–βarr1-N complex (green, βarr1; magenta, SH3; red, V2Rpp; blue, Fab30). Map contour level is 0.50. Bottom panel: The interaction interface of SH3–βarr1-N complex (green, βarr1; magenta, SH3) with density mesh (upsample map 0.69 Å/pix, contour level 0.2). Only the backbone of SH3 was modeled; βarr1 side chains are shown for illustrative purposes. See also Supplementary Video 2. **d** Western blots of M2-FLAG pull-down assay of SH3-FLAG mutants and βarr1 wild-type (WT). Data are representative of $n = 3$ independent experiments. **e** Western blots of M2-FLAG pull-down assay of SH3-FLAG wild-type (WT) and βarr1 mutants. Data are representative of $n = 3$ independent experiments.

conformational flexibility of the RT loop upon βarr1 binding. Taken together, the HDX-MS data reveal consistent qualitative trends that corroborate the disulfide trapping and cryo-EM results and support the presence of two SH3-binding regions on βarr1 and the interaction modes captured in the SH3–βarr1-CC and SH3–βarr1-N complexes.

We then compared the mechanisms of SH3 recruitment by various SH3-binding proteins and βarr1-N (Supplementary Fig. 6a-c) and βarr1-CC sites (Supplementary Fig. 6d, e)[13–17]. In all structures, SH3 utilizes its aromatic surface to engage with its binding partners (Supplementary Fig. 6). The βarr1-N site resembles the canonical polyproline-rich motif observed in Nef and ELMO1, although the polyproline sequence in βarr1 is considerably shorter (Supplementary Fig. 6a, c). In contrast, the βarr1-CC site employs non-proline residues

to recruit SH3 (Fig. 2b, Supplementary Fig. 6d, e). Non-canonical binding was previously reported for several SH3-binding proteins[18,19]. For example, endophilin A1 SH3 interacts with histidine, proline and two arginine residues in parkin Ubl (Supplementary Fig. 6b); SH3 of Sla1 endocytic protein engages with histidine and hydrophobic residues of ubiquitin (Supplementary Fig. 6e)[15,16]. βarr1 is distinguished from these examples by engaging SH3 through two mechanistically distinct binding sites, revealing an additional layer of complexity in βarr-mediated interactions.

**βarr1 drives autoinhibition relief of Src**

To understand how βarr1 activates Src, we aimed to determine the structure of βarr1 in complex with three-domain Src (SH3-SH2-SH1,

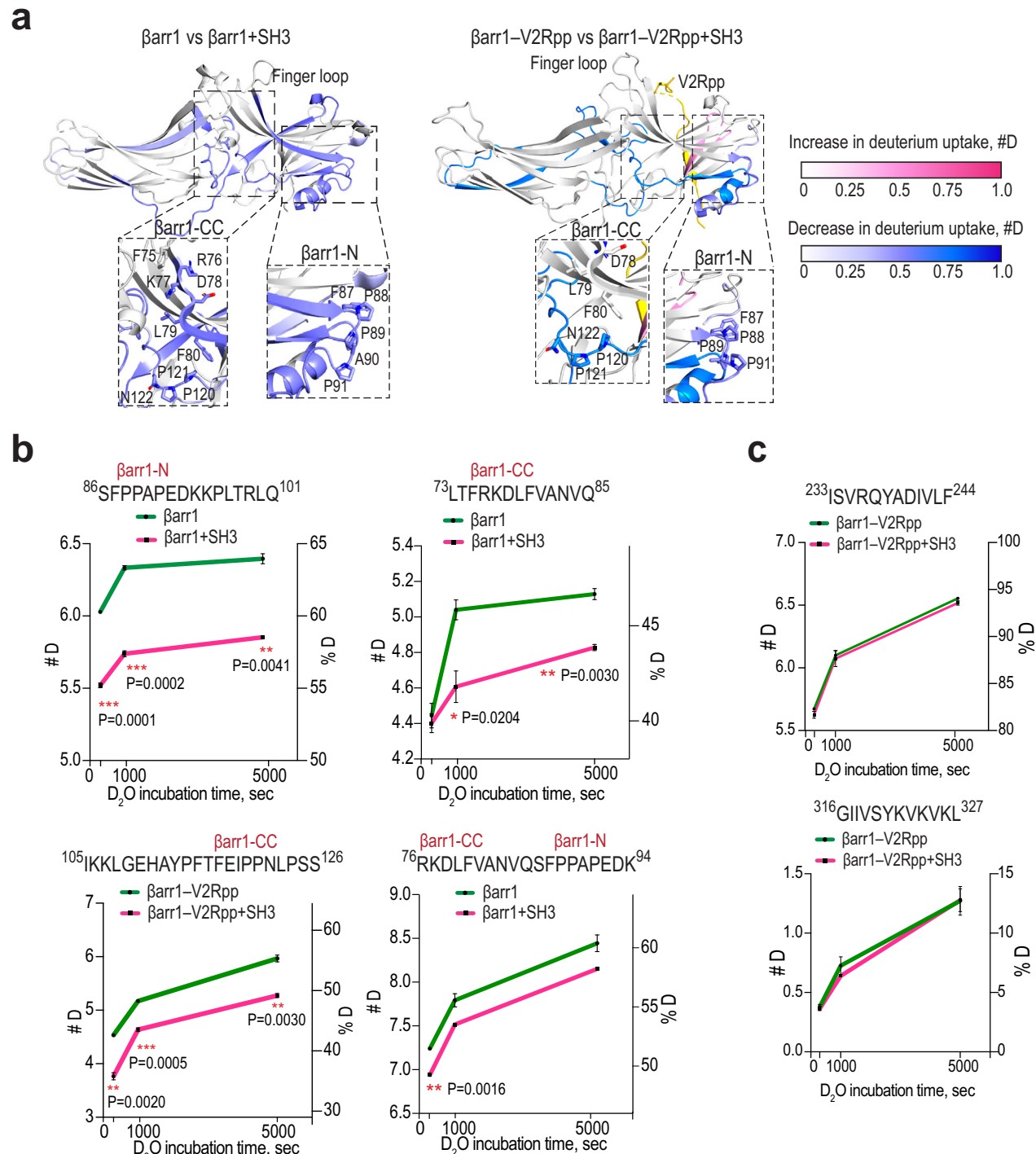

**Fig. 3 | HDX profile changes of free and V2Rpp-activated βarr1 in the presence of SH3. a** Structures of free βarr1 (PDB: 1G4M) and V2Rpp-activated βarr1 (PDB: 4JQI, grey, βarr1; yellow, V2Rpp); regions with decreased and increased deuterium uptake are shaded in purple and pink, respectively. Only regions that exhibited statistically significant differences in deuterium uptake (> 0.3 Da for βarr1; >0.36 Da for βarr1–V2rpp) are indicated. Insets: focus on the βarr1-N and βarr1-CC sites. **b** HDX changes of βarr1 peptides corresponding to βarr1-CC and βarr1-N sites upon co-incubation with SH3 (mean ± SEM, $n = 3$ technical replicates; green, βarr1 or βarr1–V2Rpp alone; magenta, βarr1 or βarr1–V2Rpp in the presence of SH3). # D and % D – deuterium uptake, Da and %, respectively. Statistical analysis was performed using Welch's t-test (*$p < 0.05$, **$p < 0.01$, ***$p < 0.001$). **c** Examples of peptides showing no changes in deuterium uptake upon co-incubation with SH3 (mean ± SEM, $n = 3$ technical replicates; green, βarr1 or βarr1–V2Rpp alone; magenta, βarr1 or βarr1–V2Rpp in the presence of SH3).

residues 83-533). We performed disulfide trapping experiments using the most promising βarr1 and Src mutants, based on the results of the βarr1–SH3 screening (Fig. 1c, d). Consistent with the findings for the isolated SH3 domain, we observed the formation of Src–βarr1 complexes at both βarr1-N and βarr1-CC sites, with βarr1 P120C (βarr1-CC)

forming the highest proportion of the cross-linked complex with Src R95C (Supplementary Fig. 7a, b). Therefore, we formed the Src_R95C–βarr1_P120C complex (hereafter referred to as Src–βarr1-CC) and determined its structure by cryo-EM (Fig. 4, Supplementary Fig. 7c, Supplementary Table 1). The final cryo-EM map has a resolution

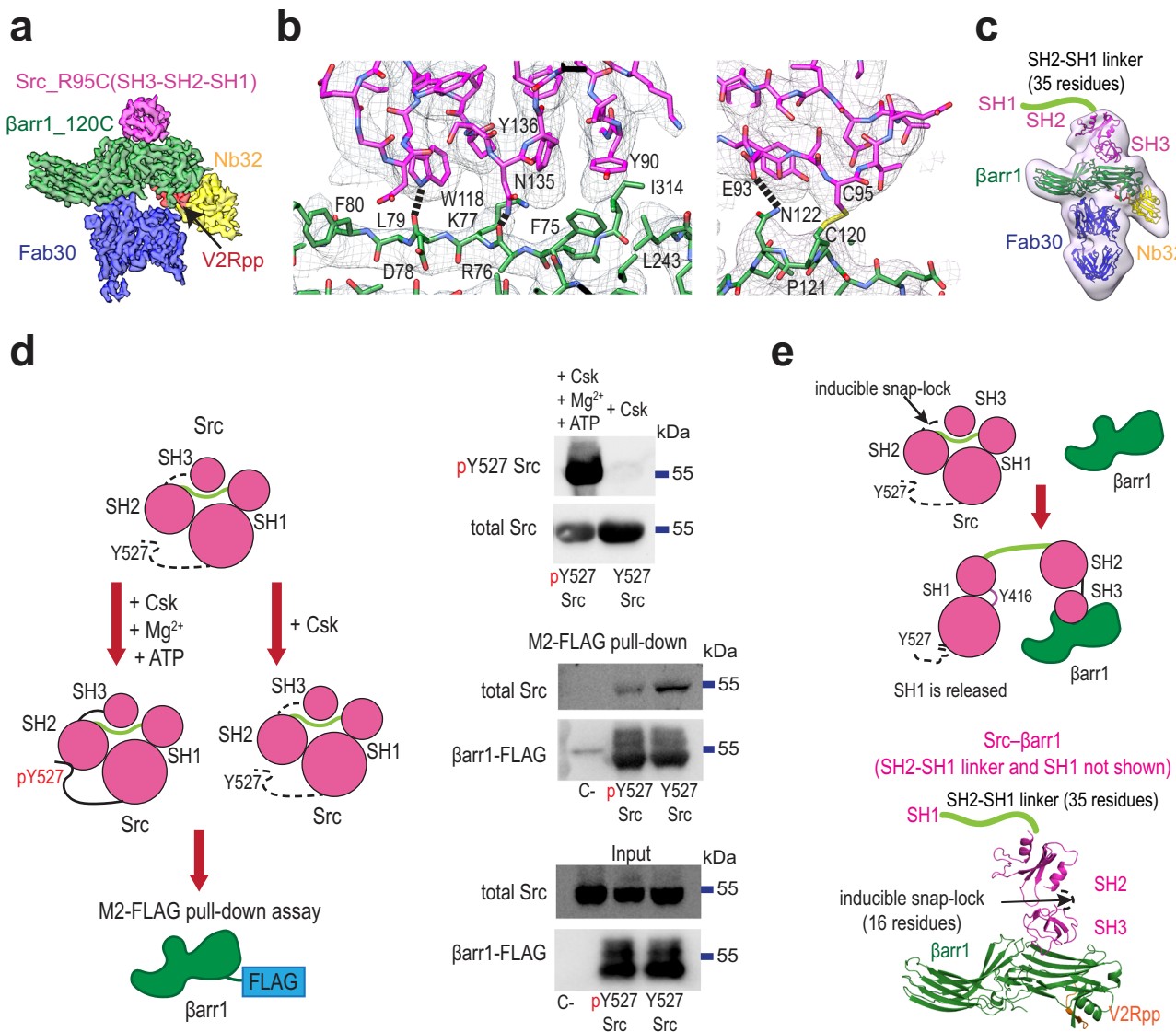

**Fig. 4 | βarr1 drives autoinhibition relief of Src. a** Cryo-EM density map of the Src–βarr1-CC complex (green, βarr1; magenta, Src(SH3); red, V2Rpp; blue, Fab30; yellow, Nb32). Map contour level is 0.65. **b** The interaction interface of Src–βarr1-CC (green, βarr1; magenta, Src (SH3 is shown) with density mesh (upsample map 0.72 Å/pix, contour level 0.15). The hydrogen bonds are shown as dashed lines. See also Supplementary Fig. 8. **c** Cryo-EM density map of the Src–βarr1-CC complex low-pass filtered to 15 Å with the fitted model (green, βarr1; magenta, Src(SH3); red, V2Rpp; blue, Fab30; yellow, Nb32). Map contour level is 0.06. **d** βarr1 preferentially binds Src lacking C-tail phosphorylation at Y527. Left panel: schematic of the assay (green, βarr1; magenta, Src). Right panel, top: representative Western blot of Src phosphorylation by Csk ($n = 4$ independent experiments). Right panel, bottom: representative Western blot of M2-FLAG pull-down assay of βarr1-FLAG and C-tail phosphorylated and C-tail unphosphorylated Src ($n = 8$ independent experiments). **e** Schematic of βarr1-mediated activation of Src (green, βarr1; magenta, Src; orange, V2Rpp). See also Supplementary Video 3.

of 3.32 Å with a clear density for the SH3 domain (Fig. 4a, Supplementary Fig. 8a-c). The superposition of SH3–βarr1-CC and the Src–βarr1-CC revealed a translational shift of 2.0-2.2 Å of the SH3 domain towards the central crest of βarr1 in the Src–βarr1-CC complex (Supplementary Fig. 8d). The orientation of SH3 in Src–βarr1-CC and the interaction network with βarr1 are similar to the SH3–βarr1-CC complex (Figs. 2b, 4b, Supplementary Fig. 8d). The 2D classes of the complex show fuzzy density around SH3 suggesting the high flexibility of the SH2 and SH1 domains (Supplementary Fig. 8e). Consistent with this observation, SH2 and SH1 were not resolved in the final map. Filtering the map to a lower resolution (15 Å) revealed the density attributable to SH2 but not to SH1 (Fig. 4c), suggesting that βarr1-CC binding to SH3 induces displacement of the SH1 domain. To determine whether a similar mechanism operates at the βarr1-N site, we formed a complex between Src R95C and βarr1 E92C (Src–βarr1-N) and collected a cryo-EM dataset. This dataset had too few usable particles to obtain a

reliable 3D reconstruction. Still, using the SH3–βarr1-N map as a reference, we aligned 2D class averages of the Src–βarr1-N complex with projections of the SH3–βarr1-N map (Supplementary Fig. 8f). This comparison revealed that the 2D projections of Src–βarr1-N closely resemble those of the SH3–βarr1-N complex, with the addition of fuzzy densities likely corresponding to the liberated SH2 and SH1 domains, mirroring the SH1 displacement observed in the Src–βarr1-CC complex.

Src activity is tightly regulated by three mechanisms of intramolecular interactions named the latch, clamp, and switch that conformationally restrict SH1, the catalytic domain of Src, and maintain its autoinhibited conformation (Supplementary Fig. 9a-c)[20,21]. Unlatching, unclamping, and switching are required for full activation of Src. The latch involves phosphorylation of C-terminal tail at Y527 and its binding to SH2 (Supplementary Fig. 9b); the clamp mediates binding of SH3 RT loop, the 3₁₀ helix and the nSrc loop to the SH2–SH1 linker and SH1

(Supplementary Fig. 9c); the switch involves the autophosphorylation of Src activation loop Y416 by another Src molecule[21]. Interestingly, superimposing the SH3–βarr1 complexes onto the crystal structure of autoinhibited Src phosphorylated at Y527 reveals steric clashes of βarr1 with the SH2 and SH1 domains, suggesting that βarr1 cannot access the SH3 domain in this conformation (Supplementary Fig. 9d). Under physiological conditions, however, the interactions between SH3, SH2, and SH1 domains are likely to be in a dynamic equilibrium between bound and unbound states, allowing the SH3 domain engaged with the SH1-SH2 linker to transiently expose its binding interface to βarr1. The affinities of these interactions are evolutionarily fine-tuned: strong enough to maintain the kinase in an autoinhibited conformation, yet sufficiently weak to respond to diverse cellular stimuli[22]. A conformation of Src lacking C-terminal phosphorylation at Y527 may further favor βarr1 accessibility, as the unphosphorylated tail forms fewer stabilizing interactions with the SH2 domain (Supplementary Fig. 9b). Supporting this, the crystal structure of unphosphorylated Src (PDB: 1Y57) shows the C-tail in a liberated state, with the C-tail-binding pocket on SH2 occupied by a sulfate ion from the crystallization buffer containing 1.3 M ammonium sulfate (Supplementary Fig. 9e). In vivo, however, unphosphorylated Src likely exists in a dynamic equilibrium between C-tail-bound and C-tail-liberated states. To determine which Src conformation is favored by V2Rpp-activated βarr1, we compared the binding of βarr1 to Src with and without C-terminal phosphorylation by pull-down assay. Phosphorylation at Y527 was performed in vitro using purified Csk, a known negative regulator of Src activity, and confirmed by western blotting (Fig. 4d). Although βarr1 is still capable of binding Y527-phosphorylated Src, it shows stronger binding to the unphosphorylated form, indicating a preference for the more dynamic, flexible conformation of Src (Fig. 4d). Previous studies have shown that the absence of C-tail phosphorylation increases the flexibility of the linker between the SH3 and SH2 domains, known as an "inducible snap lock", that may further enable the SH3 domain to accommodate not only extended polyproline sequences but also structurally diverse interaction partners[23]. Therefore, in the unphosphorylated state, enhanced flexibility in the SH3–SH2 linker appears to promote βarr1 accessibility by transiently exposing SH3 binding interface to βarr1 (Fig. 4e). As evident from the structures, βarr1 binding to SH3 RT loop and the 3₁₀ helix disrupts the clamp mechanism of Src autoinhibition and releases the SH2–SH1 linker and the SH1 domain (Fig. 2a, b, Fig. 4b, Supplementary Fig. 9a). This is further attested by the Src–βarr1-CC structure, in which high flexibility of SH1 is consistent with its unrestricted movement and the disruption of the autoinhibited state. Therefore, βarr1 causes the autoinhibition relief of Src by interacting with SH3 and disrupting the intramolecular clamp (Supplementary Video 3).

In addition to SH3, βarr1 is also known to bind the SH1 domain of Src; however, this interaction does not lead to Src activation[4]. Furthermore, in contrast to SH3, which preferentially interacts with active βarr1, SH1 shows stronger binding to the inactive form[4]. Consistent with this, our structural data show that SH1 is displaced from its position in inactive Src and does not appear to associate with V2Rpp-activated βarr1 in the conditions of our cryo-EM experiments. To gain insight into how inactive βarr1 engages SH1, we performed lysine-specific cross-linking mass spectrometry (CXMS) using purified βarr1 and SH1. CXMS provides proximity restraints between intermolecular lysine residues and between lysines and proteins N-termini (Fig. 5a, b). Two cross-linked peptides were identified between the N-terminal region of βarr1 and K356 within the C-lobe of the SH1 domain, positioned on the face opposite to the active site (Fig. 5c). To validate the CXMS data, we tested SH1 binding to different βarr1 constructs (full-length, N-domain, and C-domain) by pull-down assay. Whereas the full-length and N-domain of βarr1 bound to SH1, the βarr1 C-domain (residues 176-418) showed no detectable binding to SH1, indicating

that the interaction is mediated by the N-domain, in agreement with the cross-linking results (Fig. 5d). The structural data and the proximity restraints from cross-linking analysis suggest that βarr1 engages either SH3 or SH1, but not both simultaneously (Fig. 5e). We propose that inactive βarr1 interacts with Src through SH1; however, upon activation, this interaction appears weakened or lost, and binding shifts in favor of SH3. Further work will be needed to confirm this hypothesis.

## SH3 binding induces conformational changes in βarr1

Interestingly, βarr1 not only induces conformational changes in Src but also undergoes structural rearrangements upon Src SH3 binding (Fig. 6). The most dramatic changes are observed in the central crest region of βarr1 in the Src–βarr1-CC and SH3–βarr1-CC complexes (Fig. 6a). SH3 binding to βarr1-CC drastically changes the position of β-strand V: it shows a downward and inward movement by ~9 Å and displays a two-residue offset in its N-terminal part and a three-residue offset in its C-terminal part as compared to active βarr1 without SH3 (Fig. 6a–b). Furthermore, the middle loop in SH3–βarr1-CC adopts an intermediate conformation between the fully active and fully inactive βarr1 crystal structures (Fig. 6c). While we cannot rule out that the differences in the conformation of the middle loop may come from crystal packing constraints or the allosteric effects from binding to GPCRs[24–27], it is tempting to speculate that the larger conformational changes in β-strand V upon Src SH3 binding might affect the coupling of βarr1 to the receptor and the downstream signaling. We also observed conformational changes in βarr1 within the SH3–βarr1-N complex in proximity to the interaction interface. Specifically, the ⁴⁵PEYLKER⁵¹ loop shifts by ~6-8 Å toward SH3 and adopts a distinct conformation compared to that seen in the SH3–βarr1-CC complex and other active βarr1 structures (Fig. 6d). Furthermore, the ⁸⁷FPPAPEDK⁹⁴ motif is shifted by 4 Å toward SH3, although it remains unclear whether this change is due to disulfide cross-linking or SH3 binding (Fig. 6e). Notably, the interaction of βarr1 with Src also introduces local conformational changes in SH3 by affecting the position of the nSrc loop and converting the 3₁₀ helix into a loop (Fig. 6f).

## Physiological roles of SH3-binding sites of βarr1

We next sought to elucidate the physiological consequences of SH3 binding to βarr1-N and βarr1-CC. We generated four βarr1 mutants targeting these regions and assessed Src activation in HEK-293 βarr1/βarr2 double-knockout (CRISPR-Cas9-based, dKO) cells[28] by monitoring phosphorylation of the activation loop tyrosine Y416 following GPCR stimulation. Upon stimulation of the chimeric β2-adrenergic receptor containing the V2R tail (β2V2R) with BI-167107, βarr1 mutants exhibited a moderate reduction in βarr-dependent Src activation (Fig. 7a), indicating partial redundancy between the two Src-binding sites in this context. Notably, mutant 3, harboring combined mutations in both the βarr1-N and βarr1-CC sites (P88G_P91G_P121E_P124G), did not further decrease Src activation compared with mutant 1 (P88G_P91G) or mutant 2 (P121E_P124G). This finding suggests that βarr1–Src interactions are broadly distributed, and that the mutations tested, while disruptive, are not sufficient to fully abolish βarr-mediated Src activation.

We also tested the activation of endogenous Src in HEK-293 βarr1/βarr2 dKO cells downstream of dopamine 1 (D1R) receptor by βarr1 mutants (Fig. 7b). We chose D1R because Src is one of the key effectors of D1R signaling[29], and its activation was shown to exclusively depend on βarr[30]. While basal levels of phospho-Src were detected in all non-stimulated cells, dopamine stimulation produced a significant increase in phospho-Src in cells transfected with βarr1 WT. In contrast, most βarr1 mutants failed to respond to dopamine, indicating a reduced ability to activate Src, except for the βarr1-CC mutant P121E_P124G, which exhibited modest activation following stimulation (Fig. 7b). Taken together, both the βarr1-CC and βarr1-N sites contribute to Src activation in HEK293 cells downstream of two receptors, β2V2R and D1R.

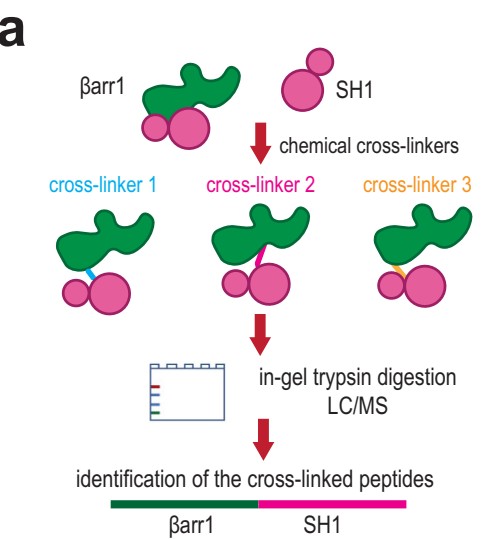

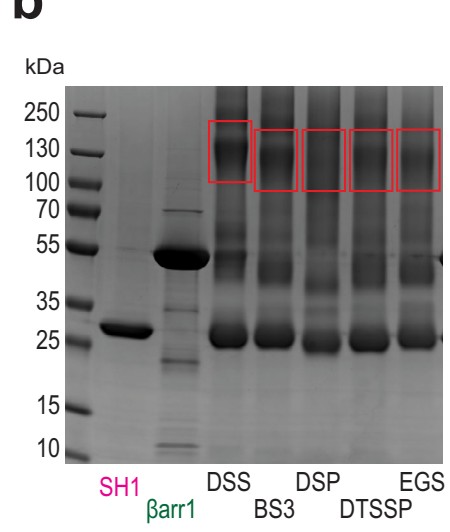

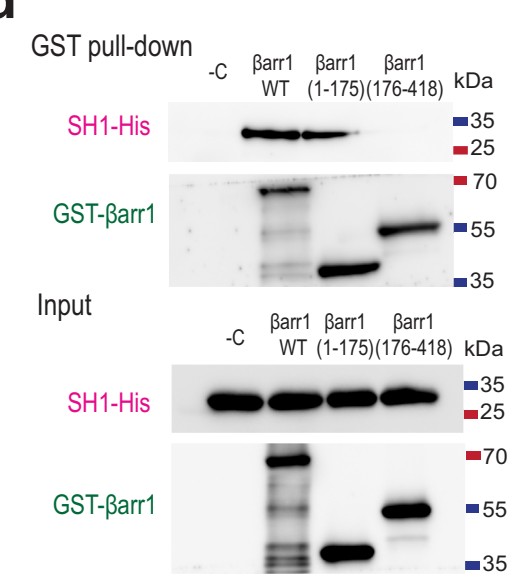

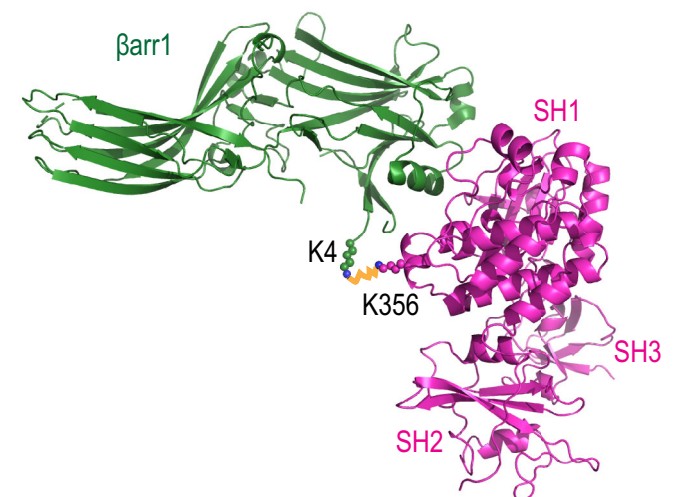

**Fig. 5 | N-domain of βarr1 interacts with SH1. a** Schematic of cross-linking mass-spectrometry (CXMS) (green, βarr1; magenta, SH1). LC-MS - Liquid Chromatography-Mass Spectrometry. **b** Chemical cross-linking of purified SH1 and βarr1, Coomassie blue gels; data are representative of $n = 3$ independent experiments. The regions of the gels excised for in-gel trypsin digestion are framed in red rectangle. DSS - disuccinimidyl suberate, BS3 -Bis(sulfosuccinimidyl) suberate, DSP – dithiobis (succinimidylpropionate), DTSSP – 3,3´-dithiobis(sulfosuccinimidylpropionate),

EGS - ethylene glycol bis(succinimidyl succinate). **c** Identified intramolecular cross-linking peptides (MaxQuant score cutoff =100). **d** Representative Western blot of GST pull-down assay of different constructs of GST-βarr1 (wild-type (WT), N-domain (1-175), C-domain (176-418)) and SH1 ($n = 8$ independent experiments). **e** Model of βarr1 (PDB: 1G4M) binding to the SH1 domain of Src (PDB: 1FMK) based on CXMS data (green, βarr1; magenta, SH1). The cross-link between K356 of SH1 and N-terminus of βarr1 is shown in yellow.

To dissect the contribution of individual residues, we generated six βarr1 mutants and assessed their ability to mediate Src activation in vitro by monitoring phosphorylation of a Src peptide substrate, using V2Rpp and Fab30 to activate βarr1. Whereas βarr1 WT increased Src activity approximately two-fold, the βarr1-N mutant (P88G_P91G), βarr1-CC triple/quadruple mutants (P121E_P124G_F75A_N122A and

P121E_P124G_F75A_I314A), as well as the double-site mutant (P88G_P91G_P121E_P124G), failed to activate Src (Fig. 7c). In contrast, CC-site mutants P121E_P124G and F80A only modestly reduced Src activation relative to WT. These results support a model in which residues P88 and P91 mediate activation via the βarr1-N site, while F75, N122, and I314 are essential for activation via the βarr1-CC site.

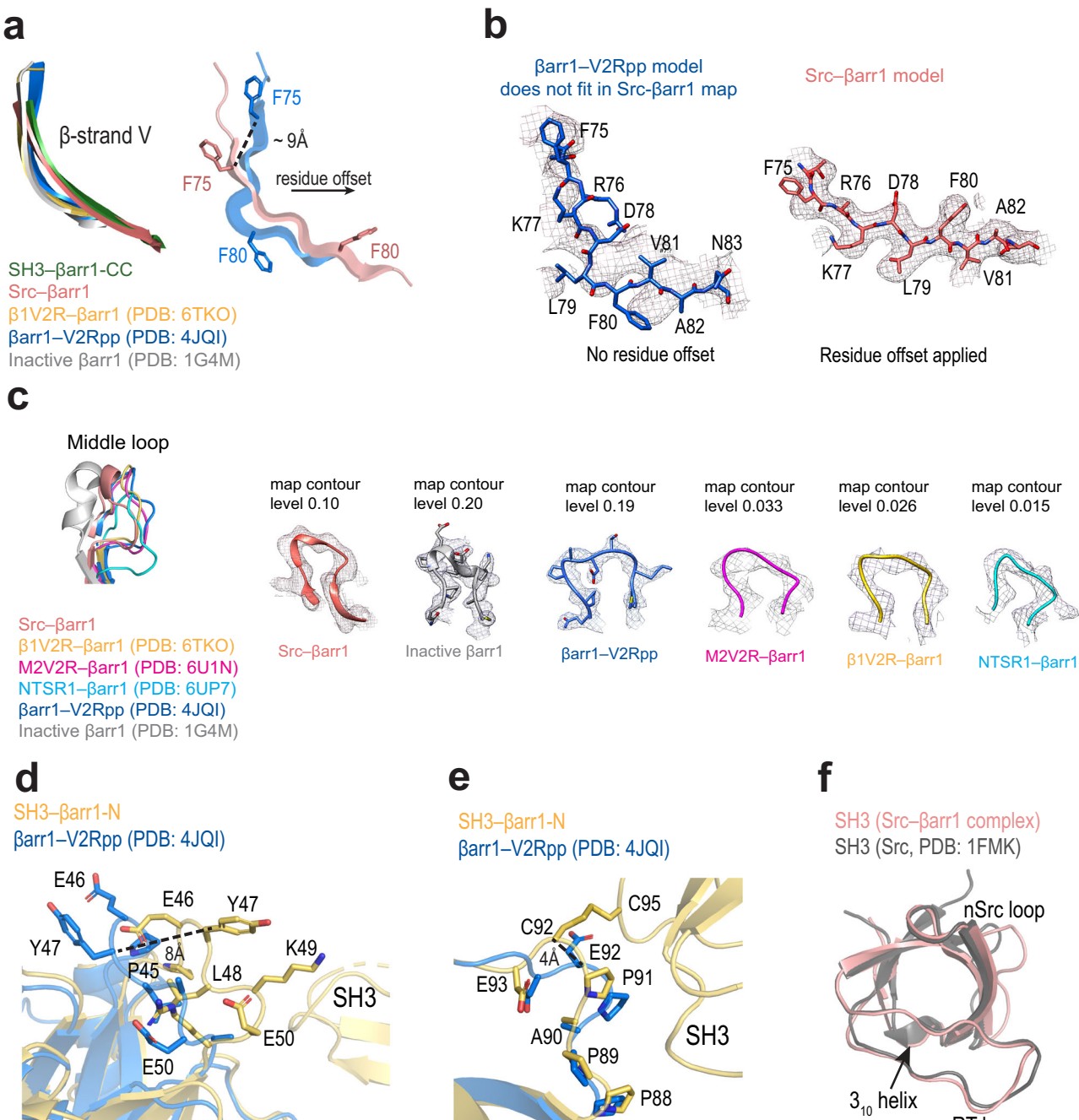

**Fig. 6 | SH3 binding induces distinct conformational changes in βarr1 and SH3.**
**a–c** Structural comparison of βarr1 in SH3–βarr1-CC (green), Src–βarr1-CC (salmon), inactive βarr1 (PDB: 1G4M, grey) and active βarr1 structures in complex with: chimeric β1AR with V2R tail (β1V2R) (PDB: 6TKO, yellow), chimeric M2 muscarinic receptor with V2R tail (M2V2R) (PDB: 6U1N, magenta), neurotensin receptor 1 (NTSR1) (PDB: 6UP7, cyan), V2Rpp–Fab30 (PDB: 4JQI, blue). **a** Focus on β-strand V. Left panel: superposition of inactive and active βarr1 structures, SH3–βarr1-CC, and Src–βarr1-CC. Right panel: residue offset in Src–βarr1-CC is shown in comparison with βarr1–V2Rpp–Fab30; the distance between Cα atoms of F75 is shown. **b** Left panel: β-strand V of the βarr1–V2Rpp–Fab30 (PDB: 4JQI, blue) does not fit into the Src–βarr1-CC density map. Right panel: the residue offset of β-strand V improves the fit into the density. The upsample map is used (0.72 Å/pix); map contour level is 0.1. **c** Focus on the middle loop. Left panel: superposition of inactive and active βarr1 structures and Src–βarr1-CC. Right panel: density around the middle loop in all respective structures (colouring is the same as in (**a**)). Structural comparison of βarr1 in SH3–βarr1-N (yellow) and active βarr1 structures in complex with V2Rpp–Fab30 (PDB: 4JQI, blue). Focus on [45]PEYLKER[51] loop (**d**) and [87]FPPAPEDK[94] motif (**e**). The distances between Cα atoms of Y47 (**d**) and E92/C92 (**e**) are shown. **f** Structural superposition of SH3 domains in Src–βarr1-CC complex (salmon) and in the crystal structure of Src (PDB: 1FMK, grey).

Interestingly, in the in vitro assay, mutation of either the βarr1-N site or selected CC-site residues completely abolished Src activation despite the presence of the other intact site. This finding indicates that in the minimal in vitro reconstitution system the two sites act cooperatively rather than independently: loss of one site compromises the ability of the other to promote activation. In cells, however, single-site mutants (all mutants downstream of β2V2R; P121E_P124G downstream of D1R) still retained partial ability to activate Src (Fig. 7a, b), suggesting that the relative contribution of each site and the extent of cooperativity between them are context-dependent, with additional scaffolding interactions and membrane confinement likely providing partial functional compensation.

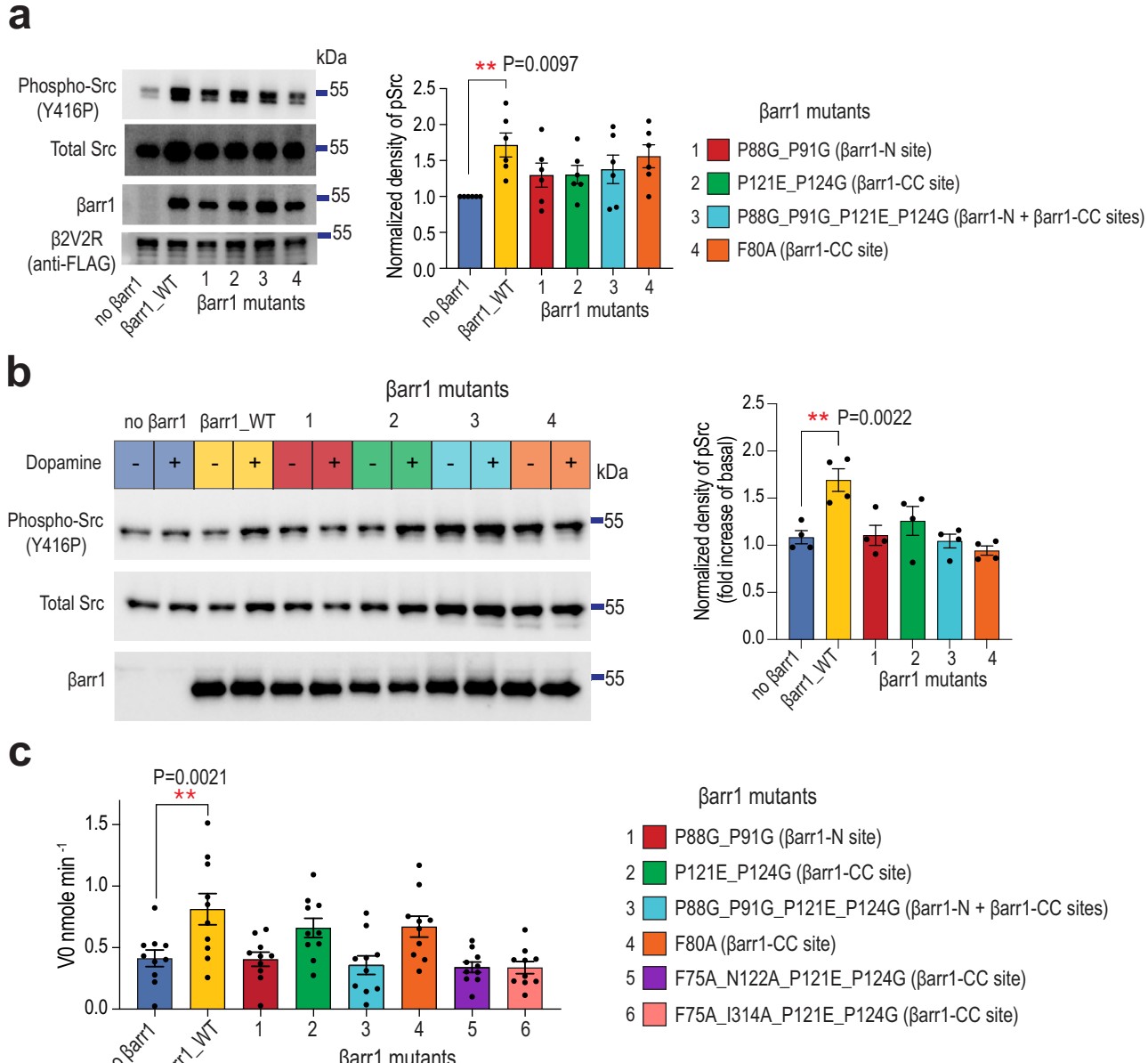

**Fig. 7 | Both βarr1-CC and βarr1-N sites mediate Src activation in vitro and in cells. a** Effect of βarr1 mutations on Src activity downstream of β2V2R. β2V2R, Src and βarr1 were transiently expressed in HEK-293 βarr1/βarr2 dKO cells. Representative Western blots and densitometry analysis of Src phosphorylation normalized to phospho-Src in control cells (no βarr1 transfected) (mean ±SEM; $n = 6$ biological replicates; one-way ANOVA with Dunnett's post hoc test; **$P < 0.01$; blue, no βarr1; yellow, βarr1 wild-type (WT); red, βarr1 P88G_P91G; green, βarr1 P121E_P124G; cyan, βarr1 P121E_P124G; orange, βarr1 F80A). **b** Effect of βarr1 mutations on endogenous Src activity downstream of D1R. D1R and βarr1 were transiently expressed in HEK-293 βarr1/βarr2 dKO cells. Representative Western blots and densitometry analysis of Src phosphorylation normalized to phospho-Src in unstimulated cells (mean ±SEM; $n = 4$ biological replicates; one-way ANOVA with Dunnett's post hoc test; **$P < 0.01$) βarr1 mutants numbering and colouring is the same as in (**a**). **c** Initial velocity of optimal Src peptide phosphorylation by Src (V0) alone or in the presence of βarr1 WT or βarr1 mutants activated by V2Rpp and Fab30. (mean ±SEM; $n = 10$ independent experiments; one-way ANOVA with Dunnett's post hoc test; **$P < 0.01$; blue, no βarr1; yellow, βarr1 wild-type (WT); red, βarr1 P88G_P91G; green, βarr1 P121E_P124G; cyan, βarr1 P121E_P124G; orange, βarr1 F80A; purple, βarr1 F75A_N122A_P121E_P124G; salmon βarr1 F75A_I314A_P121E_P124G).

We hypothesized that binding of the Src SH3 domain to the central crest region of β-arrestin1 (βarr1-CC), along with the resulting conformational changes in βarr1, could impair its engagement with the transmembrane core of GPCRs (Fig. 8a). We, therefore, evaluated whether excess of SH3 would interfere with the core coupling of βarr1 to the phosphorylated V2R in membranes. We labeled the finger loop of βarr1 V70C with an environment-sensitive bimane fluorophore (mBr) and measured the changes in fluorescence emission spectra with and without SH3 (Fig. 8b). Upon stimulation of the phosphorylated V2R with arginine-vasopressin peptide (AVP), the finger loop of βarr1 V70C-mBr binds to the receptor core, leading to a ~50% increase in bimane fluorescence. The presence of SH3 significantly reduces this effect, suggesting that SH3 binding to βarr1 sterically hinders the core coupling of βarr1 to V2R. Similar results were obtained with the purified phosphorylated chimeric M2 muscarinic receptor with V2R tail (M2V2R) reconstituted in lipid nanodiscs (Fig. 8c). These data suggest that high local concentration of Src can directly influence the conformational equilibrium of GPCR–βarr1 complexes by disrupting the fully engaged "core" conformation (Fig. 8d), which, in turn, may impair receptor desensitization and hinder the termination of G

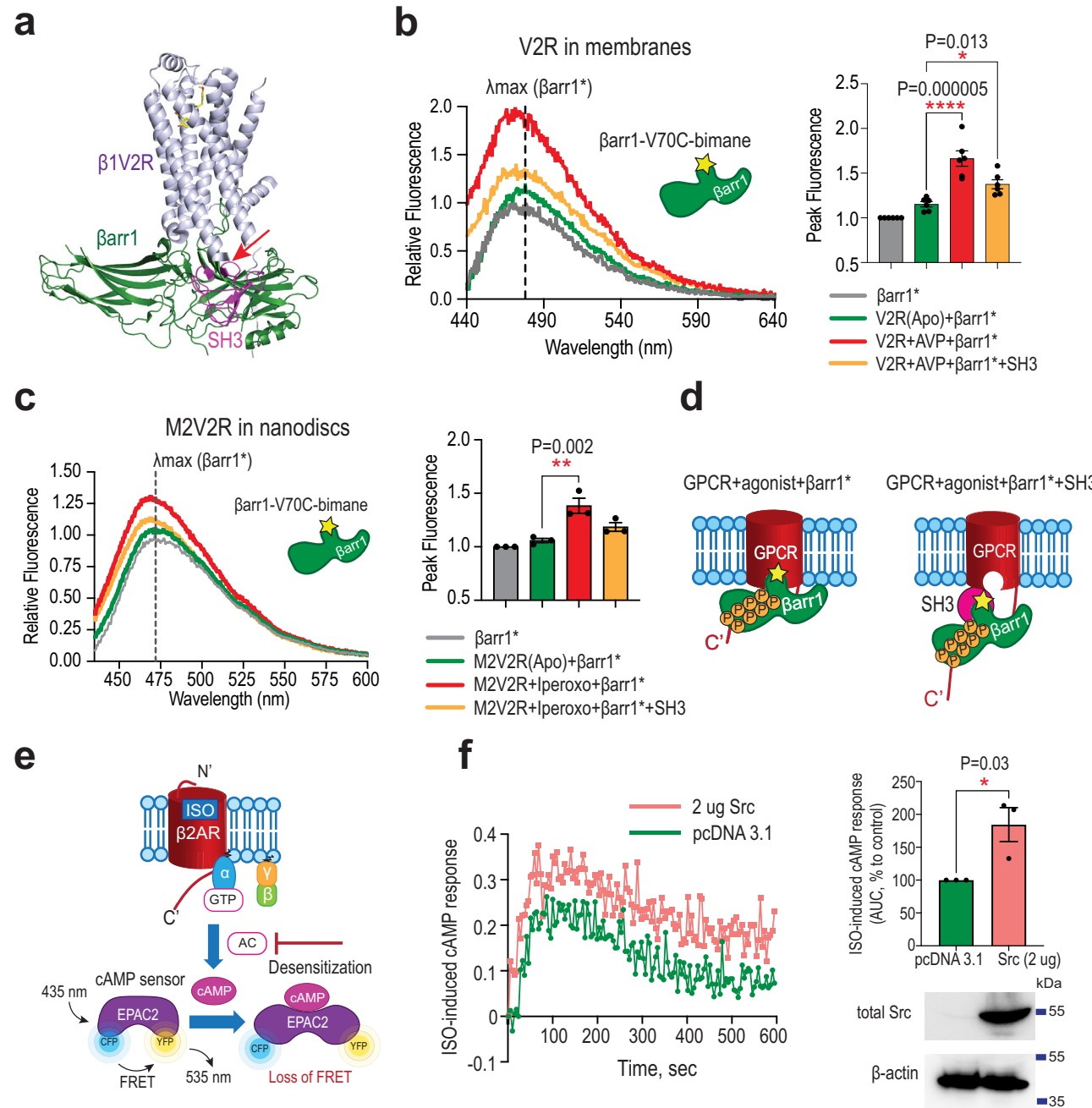

**Fig. 8 | Excess of Src SH3 interferes with the core coupling of βarr1 to GPCRs and overexpression of Src affects desensitization of β2AR. a** Structural superposition of Src–βarr1-CC (SH3, magenta; βarr1, green) with the β1V2R–βarr1 complex (β1V2R, slate; PDB: 6TKO), cartoon representation. The clashing region is indicated with the red arrow. **b** Core coupling of βarr1 finger loop bimane to phosphorylated V2R in membranes with and without SH3. Left panel: representative experiment of βarr1 finger loop bimane fluorescence. Right panel: peak fluorescence of βarr1 finger loop bimane (mean ±SEM; $n = 6$ independent experiments; one-way ANOVA with Dunnett's post hoc test; *$P < 0.05$, ****$P < 0.0001$; grey, βarr1 alone; green, V2R(Apo)+βarr1; red, V2R + AVP+βarr1; yellow, V2R + AVP+βarr1 + SH3). **c** Core coupling of βarr1 finger loop bimane to the phosphorylated chimeric M2V2R reconstituted in lipid nanodiscs. Left panel: representative experiment of βarr1 finger loop bimane fluorescence. Right panel: peak fluorescence of βarr1

finger loop bimane (mean ± SEM; $n = 3$ independent experiments; one-way ANOVA with Dunnett's post hoc test; **$P < 0.01$; grey, βarr1 alone; green, M2V2R(Apo) +βarr1; red, M2V2R+Iperoxo+βarr1; yellow, M2V2R+Iperoxo +βarr1 + SH3). **d** Schematic of SH3-mediated disruption of βarr1 core coupling to GPCRs (red, GPCR; green, βarr1; magenta, SH3). **e** Schematic of the real-time Exchange Protein Activated by cAMP 2 (Epac2)-Fluorescence Resonance Energy Transfer (FRET) cAMP accumulation assay. cAMP binding leads to increased CFP–YPF distance and reduces FRET efficiency. **f** Left panel: representative kinetic profiles from HEK293 cells stably expressing β2AR and Epac2 and transiently transfected with 2 ug Src (salmon) or control DNA (green). Right panel, top: area under the curve (AUC) of FRET responses relative to control (100%). Right panel, bottom: representative Western blots of cell lysates (mean ± SEM; $n = 3$ biological replicates; two-tailed Welch's $t$ test; *$P < 0.05$).

protein–mediated signaling. To test this hypothesis in a cellular context, we monitored real-time cAMP dynamics in HEK293 cells stably expressing the β2-adrenergic receptor (β2AR) using the live-cell FRET-based ICUE2[31] biosensor, which consists of a fusion between cyan and

yellow fluorescent proteins (CFP-Epac2-YFP) (Fig. 8e). Stimulation of Gαs-coupled β2AR with isoproterenol triggers a rapid peak in cAMP levels, followed by a gradual decline as a result of β2AR desensitization. Overexpression of Src led to increased agonist-induced cAMP

accumulation whereas the decline of cAMP accumulation was delayed compared to control cells, indicating that β2AR desensitization is affected (Fig. 8f).

## Discussion

GPCRs convert extracellular signals into cellular responses via two major transducers, G proteins and βarrs, which in turn interact with their downstream effectors. The rapid development of cryo-EM has greatly advanced our understanding of how GPCRs recruit signal transducers. In contrast, little is known about how the transducers regulate their effector enzymes. Protein interactions within signaling cascades are inherently weak and transient, which is crucial for the dynamic regulation of cellular processes. Capturing dynamic and transient protein complexes has long posed a significant challenge in structural biology, often requiring stabilization strategies such as genetic fusions, chemical cross-linking, disulfide trapping, or extensive mutagenesis to obtain structural information that would otherwise be unattainable[32–34]. Among these methods, disulfide trapping offers distinct advantages: it minimizes non-specific artifacts compared to chemical cross-linking and has a lower risk of disrupting protein folding or expression compared to genetic fusion approaches. Using disulfide trapping, we determined the high-resolution structures of a βarr-effector signaling hub, βarr1 in complex with tyrosine kinase Src and isolated SH3 bound to two distinct sites of βarr1. The structural data reveal a distinct strategy by which βarr1 recruits the Src SH3 domain. βarr1 binds the aromatic surface of SH3 using two distinct sites, each with its own binding mechanism. The βarr1-N site is a short polyproline motif, whereas the βarr1-CC site is formed by the non-proline residues in the β-strand V and the lariat loop, as well as two proline residues.

The ability of βarr1 to engage its effector proteins through two distinct sites using different binding mechanisms highlights a sophisticated level of modularity and flexibility in signaling scaffolds. This dual-site architecture likely plays a central role in the remarkable functional versatility of βarr1 across diverse cellular contexts. Several key implications emerge from this mode of engagement. First, this configuration could facilitate the simultaneous or sequential recruitment of effectors, enabling the assembly of multi-component signaling complexes or temporally orchestrated signaling cascades. In the context of Src, βarr-mediated conformational activation at two distinct sites may facilitate conditions compatible with Src trans-autophosphorylation by locally increasing the effective concentration of kinase molecules. In proteins containing multiple SH3 domains, dual-site engagement can enhance avidity through classical multivalency; for Src, which contains a single SH3 domain, proximity-driven rebinding likely fulfills an analogous functional role. Second, the dual-site architecture might contribute to the spatiotemporal compartmentalization of Src signaling[35], with one site functioning predominantly at the plasma membrane and the other in endosomal compartments. Such spatial regulation could underlie distinct modes of Src activation (transient vs. sustained) and lead to different signaling outcomes, such as rapid cytoskeletal rearrangement or longer-term transcriptional responses. Third, although Src contains a single SH3 domain and can engage only one βarr1 site at a time, the presence of two spatially proximal SH3-binding motifs within βarr1 is likely to enhance Src recruitment through avidity-like effects. Rather than functioning as redundant, independent docking sites, the two weak-affinity sites increase the effective local concentration of SH3 ligands and promote rapid rebinding following dissociation, thereby stabilizing what would otherwise be transient interactions. This model explains why mutation of either site abolishes Src activation in the minimal in vitro system, where cooperative rebinding is essential, while single-site mutants retain partial activity in cellular contexts, where additional scaffolding interactions and membrane confinement can partially compensate. Thus, the dual-site architecture of βarr1

appears optimized to enhance the kinetics and stability of Src engagement rather than to provide simple redundancy. Finally, different upstream signals may induce conformational states of βarr1 that preferentially expose one Src-binding site over the other. While our data show that signaling downstream of β2V2R and D1R did not display a clear bias toward either site, this may not be the case for other GPCRs or in response to different ligands, presenting an exciting direction for future research. Furthermore, post-translational modifications of βarr1 could influence the accessibility or affinity of each site, allowing βarr1 to function as a dynamic molecular gate with context-dependent control over Src activation. Beyond Src, this dual-recognition strategy may represent a broader mechanism by which βarr1 engages other SH3 domain-containing proteins. Elucidating the biological consequences of the dual-site recognition mechanisms of βarr1 opens exciting avenues for exploring its complex regulatory roles in cellular signaling.

Src activation by SH3/SH2-binding proteins is well established: binding to these domains disrupts the intramolecular interactions that constrain the kinase activity[36–42]. Prior studies suggested that βarr1 activates Src via SH3 domain interaction[3,4]. Using site-specific fluorescence labeling of Src, Yang et al. showed the disruption of SH3/SH1 and SH3/SH2 interactions in the presence of β2AR-βarr1 complex[3]. However, in the absence of structural data, it remained unclear whether these effects result from βarr1's scaffolding or its direct activation of Src. Here, we present the structural evidence that βarr1 functions as an active regulatory protein rather than a passive scaffold and elucidate the precise molecular mechanism of βarr1-mediated allosteric activation of Src (Fig. 4, Supplementary Video 3). At both sites, βarr1 interacts with the aromatic surface of SH3, which is critical for maintaining the autoinhibited conformation of Src. This interaction disrupts the autoinhibitory interactions, liberating the catalytic domain of Src, as observed in the structure of the Src–βarr1-CC complex.

Notably, Src SH3 and βarr1 exhibit reciprocal effects: βarr1 induces conformational changes in Src promoting its activation, while Src SH3 simultaneously triggers significant structural rearrangements in βarr1. The conformational changes in the central crest region of βarr1 upon SH3/Src binding may affect βarr1 interactions with GPCRs and other signaling partners (Fig. 6). Our results demonstrate that an excess of SH3 interferes with βarr1 coupling to the receptor core (Fig. 8b–d). Furthermore, in HEK293 cells overexpressing Src, we observed elevated agonist-induced cAMP accumulation, indicating impaired β2AR desensitization (Fig. 8e, f). Although the βarr1–Src interaction is likely transient, a high local concentration of Src may shift the conformational equilibrium of GPCR–βarr1 complexes toward a "tail-only" engagement, which permits sustained G-protein signaling. This mechanism is reminiscent of sustained G-protein signaling from internalized GPCRs via megaplex assemblies, in which GPCRs simultaneously engage both G-proteins through its core region and βarr1 through its phosphorylated C-terminal tail[43,44]. Future structural studies on the GPCR–βarr–Src axis will be crucial to define the conformational states of the complex and assess their physiological relevance. A promising direction is to test the implications of this mechanism in cancer models, where Src is frequently overexpressed[45], to understand its role in disease progression and signaling dysregulation. Our results suggest that Src overexpression may disrupt receptor desensitization.

βarrs, originally discovered as proteins mediating GPCR desensitization[46,47], are now commonly recognized as signal transducers[2–4,30]. Nevertheless, some controversy remains as to whether βarrs are just modulators of G-protein-mediated signaling events[48] or rather bona fide signal transducers in their own right. Our cryo-EM structures support the latter: βarr1 recruits Src and allosterically activates the enzyme by disrupting its autoinhibitory intramolecular interactions. This may represent a general mechanism by which βarrs mediate signal transduction across diverse effectors. Beyond allosteric activation, βarrs might also facilitate signaling by positioning Src near

its substrates, analogous to the arrangement seen in focal adhesion kinase (FAK)−Src signaling complexes[49]. Taken together, our study demonstrates how βarrs orchestrate cellular signaling downstream of GPCRs and highlights avenues for exploring their regulatory roles in both physiological and pathological contexts.

## Methods

### Bacterial strains
Escherichia coli strain DH5α (C2987, New England Biolabs) was used for plasmid propagation. Protein expression was done in E. coli BL21 derivative strain (C3010, New England Biolabs).

### Cell culture
Human embryonic kidney CRISPR-Cas9-based βarr1/βarr2 double knock-out cell line (HEK-293 βarr1/βarr2 dKO)[28] was maintained in Gibco Minimum Essential Media (MEM) supplemented with 1% penicillin-streptomycin and 10% (v/v) fetal bovine serum at 37 °C and 5% $CO_2$. Expi-293F (Invitrogen, A14527) and tetracycline-inducible Expi293F cells (Invitrogen, A39241) were grown in suspension in Expi293 Expression Medium (Gibco) at 37 °C and 8% $CO_2$.

### Molecular biology
Bacterial expression constructs for wild-type (residues 2-418) and a minimal cysteine (C59A, C125S, C140I, C150V, C242V, C251V, and C269S) and truncated (βarr1−MC-393) variants of rat βarr1[50], Fab30[51], and Nb32[11] have been previously reported and functionally verified. Plasmid pET28a-3D-Src expressing SH3, SH2 and SH1 domains of chicken Src (residues 83-533), plasmids expressing YopH phosphatase and Csk were generous gifts from Prof. John Kuriyan. For structural studies, a Src-5Cys mutant (C185S, C238S, C245S, C277S, C400S) was generated. Bacterial expression constructs for Src SH1 domain (250-536) and for βarr1 C-domain (176-418) were synthesized by GenScript. Plasmid expressing the SH3 domain of Src (residues 83-141) was designed by inserting a C-terminal FLAG-tag and a stop codon in pET28a-3D-Src. Mammalian expression constructs for human FLAG-M2-muscarinic receptor (M2R) with C-terminal sortase ligation consensus sequence (LPETGGH) and 6×His-tag[50], chimeric FLAG-β2-adrenergic receptor with C-terminal tail of vasopressin 2 receptor (β2V2R)[52], SNAP-vasopressin 2 receptor (V2R)[43], HA-tagged βarr1[9], and human Src[9] have been previously reported and functionally verified. Mammalian expression construct for HA-dopamine 1 receptor (D1R) was a gift from Prof. Marc Caron. Mutations were introduced using QuickChange II site-directed mutagenesis kit (Agilent) and verified by Sanger sequencing. Oligonucleotides used for site-directed mutagenesis are listed in Supplementary Table 5.

### Protein expression and purification
The Src construct (residues 83-533) without the N-terminal disordered region was used for the structural and in vitro studies[53]. Src was co-expressed in E. coli with YopH phosphatase and purified by immobilized metal ion affinity chromatography and anion exchange chromatography. The SH3 domain of Src (residues 83-141) and its variants and the SH1 domain of Src (251-533) were expressed in E. coli and purified by immobilized metal ion affinity chromatography followed by size exclusion chromatography (SEC). Wild-type βarr1[54] and its variants were expressed in E. coli and purified using GST affinity chromatography followed by anion-exchange chromatography. Fab30[55] was expressed in E. coli and purified using protein A affinity chromatography. Nb32[11] was expressed in E. coli and purified using immobilized metal ion affinity chromatography followed by SEC. Csk[56] was expressed in E. coli and purified using GST affinity chromatography followed by anion-exchange chromatography. M2R containing a C-terminal sortase ligation consensus (LPETGGH)[25,50] was stably expressed in tetracycline-inducible Expi293F cells (Invitrogen, A39241), solubilized using 1% n-dodecyl-β-D-maltopyranoside (Anatrace), 0.05% cholesterol hemisuccinate and purified by M1 FLAG affinity chromatography. The synthetic phosphopeptide GGG-V2Rpp (GGG-ARGRpTPPpSLGPQDEpSCpTpTApSpSpSLAKDTSS) was ligated to the C-terminus of M2R using sortase, and the monomeric chimeric M2V2R was collected by SEC[50]. M2V2R was reconstituted in lipid nanodiscs using Membrane Scaffold Protein 1, variant D1 with E3 helices added (MSP1D1E3) and a 3:2 molar ratio of phosphatidylcholine with phosphatidylglycerol. Detergent was removed using Biobeads (BioRad)[25].

### Preparation of crude membranes with V2R and G-protein coupled receptor kinase 2 (GRK2)
Expi-293 cells grown in suspension were co-transfected with plasmids expressing the human V2R and the membrane targeted form of GRK2 to promote receptor phosphorylation. Transfected cells were grown in suspension for 48 hours and thereafter were stimulated with AVP (100 nM) for 30 min at 37 °C followed by harvesting by centrifugation at 4 °C. Cell pellets were washed with cold Hanks' Balanced Salt Solution, snap frozen in liquid nitrogen and stored at −80 °C prior to preparation of crude membranes[57]. Frozen cells were dounce-homogenized in cold homogenization buffer (20 mm Tris-HCl pH 7.4), containing EDTA-free protease inhibitor cocktail (Roche) and PhosSTOP phosphatase inhibitor cocktail (Roche). Following differential centrifugation, the P2 microsomal membrane fraction was resuspended in cold buffer (50 mm Tris-HCl, 150 mm NaCl, 12.5 mm MgCl2, 0.2% BSA, and 10% glycerol, pH 7.4), supplemented with protease inhibitor cocktail and PhosSTOP phosphatase inhibitor cocktail, aliquoted, and stored at −80 °C until use.

### Disulfide trapping
βarr1−MC-393 mutant (30 μM) was incubated with 3-fold molar excess of V2Rpp (ARGRpTPPpSLGPQDEpSCpTpTApSpSpSLAKDTSS), 1.5 molar excess of Fab30, and 2-fold molar excess of SH3 mutant or Src mutant in 20 mM HEPES 7.5, 150 mM NaCl buffer for 30 minutes at room temperature. The disulfide trapping reactions were initiated by 1 mM $H_2O_2$. After 1-hour incubation at room temperature the reactions were terminated by addition of 4x Laemmli sample buffer (without β-mercaptoethanol) (BioRad), subjected to SDS-PAGE, visualized by Ready Blue Coomassie stain (Sigma-Aldrich) and quantified by ImageJ v1.52a. The band of SH3−βarr1 complex (~60 kDa) or Src−βarr1 (~130 kDa) was normalized to the total density of all bands in each sample. To test the effect of V2Rpp and Fab30 on formation of disulfide trapped complexes, the reactions were performed as described above; in reactions without V2Rpp and Fab30 the equivalent amount of buffer was added. The samples were subjected to SDS-PAGE and Western blotting, detected by rabbit polyclonal A1CT antibody generated in Lefkowitz lab[58] (1:5000) and HRP-conjugated ECL Rabbit IgG (NA9340, Amersham, RRID:AB_772191) (1:5000) for βarr1 and HRP-conjugated Anti-6X His-tag antibody (1:2000) (ab1187, Abcam, RRID: AB_298652) for SH3.

### Isothermal titration calorimetry (ITC)
ITC measurements were performed using the MicroCal PEAQ-ITC system (Malvern). Purified βarr1−V2Rpp and SH3 were dialyzed in 20 mM HEPES, pH 7.5, 100 mM NaCl. 75 μM of βarr1−V2Rpp was loaded into the sample cell and 750 μM of SH3 into the injection syringe. The system was equilibrated to 25 °C. Titration curves were initiated by a 0.4 μL injection from syringe, followed by 2.0 μL injections (at 180 s intervals) into the sample cell. During the experiment, the reference power was set to 7 μcal·s-1 and the sample cell was stirred continuously at 750 rpm. Raw data, excluding the peak from the first injection, were baseline corrected, peak area integrated, and normalized. Data were analyzed using MicroCal Origin software to obtain thermodynamic parameters of binding and association constant (Ka=1/Kd).

## Formation of SH3–βarr1 and Src–βarr1 complexes

The βarr1–MC-393 mutant (100 μM) was incubated with 2-fold molar excess of the SH3 mutant or Src mutant for 30 minutes at room temperature, followed by disulfide trapping as described above. SH3–βarr1 was separated from unbound βarr1 by immobilized metal ion affinity chromatography using Talon resin (Takara Bio). Src–βarr1 was separated from unbound βarr1 and Src by SEC. SH3–βarr1 and Src–βarr1 complexes were then incubated with 3-fold molar excess of V2Rpp, 1.5-fold molar excess of Fab30, 2-fold molar excess of Nb32 (for SH3–βarr1-CC and Src–βarr1-CC) for 30 minutes at room temperature. The complexes were subjected to SEC on a Superdex 200 Increase column (Cytiva Life Sciences) in 20 mM HEPES 7.5, 150 mM NaCl buffer. Peak fractions were concentrated to 6-8 mg/ml using Vivaspin® 6 column with molecular weight cut-off of 30,000 kDa (Sartorius).

## Cryo-EM grid preparation and data acquisition

In the preliminary cryo-EM experiments, we observed that the complexes exhibited strong preferred orientation in vitreous ice. To alleviate the preferred orientation problem, 8 mM CHAPSO was added to the sample immediately before vitrification[59]. The sample was applied to glow-discharged 300-mesh holey-carbon grids (Quantifoil R1.2/1.3, Electron Microscopy Sciences) or gold grids (Ultrafoil R0.6/1.0, Electron Microscopy Sciences) and vitrified using a Vitrobot Mark IV (Thermo Fisher Scientific) at 4 °C and 100% humidity. The data were collected on a Titan Krios transmission electron microscope (Thermo Fisher) operating at 300 kV equipped with a K3 direct electron detector (Gatan) in counting mode with a BioQuantum GIF energy filter (slit width of 20 eV) at a magnification of ×81,000 corresponding to a pixel size of 1.08 Å at the specimen level. 60-frame Videos with a dose rate of ~15 electrons per pixel per second and a total accumulated dose of ~53-60 electrons per Å$^2$ were collected using the Latitude-S (Gatan) single-particle data acquisition program. The nominal defocus values were set from −0.8 to −2.5 μm.

## Cryo-EM data processing

Videos were subjected to beam-induced motion correction using Patch Motion Correction in CryoSPARC v4.0.1[60] followed by determination of Contrast transfer function (CTF) parameters in Patch CTF. Micrographs with CTF fit better than 3.5 Å were used for further analysis. Particles manually selected from 15 micrographs were used to train a model in the particle picking tool Topaz v0.2.5a[61]. The trained model was used to pick particles in all micrographs generating 200,270 particle projections for the SH3–βarr1-CC complex and 760,233 particle projections for the SH3–βarr1-N complex. The particles were rescaled to the pixel size of 1.3824 Å for further processing. For the SH3–βarr1-CC complex, the particle stack was subjected to 2D classification, Ab Initio model generation, non-uniform refinement and local refinement with a mask excluding the constant (CL/CH1) region of Fab30 (FabCR) in CryoSPARC. Particles were then imported to RELION v3.1[62] and subjected to 3D classification without alignment with a mask excluding the constant (CL/CH1) region of Fab30 (FabCR). Classes showing a clearly defined density for the SH3 domain (118,020 particles) were re-imported to CryoSPARC and subjected to local refinement with a mask on FabCR and a fulcrum on the SH3–βarr1 interface (Supplementary Fig. 3a). The resulting map was then processed by spIsoNet to improve map isotropy[63]. The final map has a global resolution of 3.47 Å at a Fourier shell correlation of 0.143 (Supplementary Fig. 4c). For the SH3–βarr1-N complex, the particles picked with Topaz were subjected to 2D classification and the best 2D classes were used in template picking (5,607,258 particles). A subset of particles (1,000,000) was used to generate six Ab Initio classes. All particles were then subjected to four rounds of heterogeneous refinement. The resulting particle stack (345,529) was used in non-uniform refinement and local refinement with a mask excluding FabCR (Supplementary Fig. 3b). The resulting map was then subjected to

spIsoNet to improve map isotropy[63]. The final map has a global resolution of 3.34 Å at a Fourier shell correlation of 0.143 (Supplementary Fig. 4d). For the Src–βarr1-CC complex, the particles picked with Topaz (94,196 particles) were subjected to 2D classification and the best 2D classes were used in template picking (9,770,378 particles). The particles were rescaled to the pixel size of 1.44 Å for further processing. A subset of particles (1,000,000) was used to generate three Ab Initio classes. All particles were then subjected to three rounds of heterogeneous refinement, followed by non-uniform refinement and local refinement with a mask excluding FabCR. The resulting particle stack (692,850) was subjected to 3D classification without alignment in CryoSPARC and then in RELION with a mask excluding FabCR. Finally, the best class with 140,156 particles was subjected to another round of local refinement in CryoSPARC, generating a map with a global resolution of 3.32 Å (Supplementary Fig. 7c, Supplementary Fig. 8b). For the Src–βarr1-N complex, the particles picked with Topaz (61,962 particles) were subjected to 2D classification and matched with the 2D projections of the SH3–βarr1-N map using cluster selection mode in Reference Based Auto Select 2D module in CryoSPARC.

## Model building and refinement

Cryo-EM maps for SH3–βarr1 complexes were post-processed using DeepEMhancer[64] to improve the interpretability of maps, specifically in the βarr1 loops and at the SH3–βarr1 interface. Since the overall resolution of maps was better than 4 Å, highRes deep learning model was used during DeepEMhancer processing of maps. The initial models of the SH3–βarr1-CC complex and Src–βarr1-CC complex were built manually by fitting the crystal structures of SH3 (PDB: 2PTK), βarr1–V2Rpp–Fab30–Nb32 complex (PDB code: 6NI2) into the experimental electron densities using UCSF Chimera v1.15[65]. The structures were refined by combining manual adjustments in Coot v0.9.8.3[66] and ISOLDE[67] in UCSF ChimeraX 1.6.1[68], followed by real-space refinement in PHENIX v1.20.1-4487[69] with Ramachandran, rotamer, torsion, and secondary structure restraints enforced. For the SH3–βarr1-N complex, the crystal structure of the βarr1–V2Rpp–Fab30 complex (PDB code: 4JQI) was first built manually into the density and refined similarly to the SH3–βarr1-CC complex. After the refinement, SH3 (PDB: 2PTK) was docked into the density guided by the position of the disulfide bond between βarr1_E92C and SH3_R95C. The rigid body fitted SH3–βarr1-N model generated above was then subjected to molecular dynamics flexible fitting (MDFF) using Cryo fit[70], integrated in Phenix software suite and Namdinator[71] followed by further iterative rounds of model building and real space refinement in Coot v0.9.8.3[66] and PHENIX v1.20.1-4487[69]. Due to weak density for the SH3 domain in the SH3–βarr1-N complex, only its main chain was modeled. The models were validated with MolProbity v4.5.1[72]. Refinement statistics are given in Supplementary Table 1.

## Hydrogen-deuterium exchange mass-spectrometry

To map the βarr1 interface involved in SH3 binding, βarr1–MC-393 (20 μM) or βarr1–MC-393–V2Rpp (20 uM βarr1; 100 uM V2Rpp) were incubated with 5-fold molar excess of SH3 in the 20 mM HEPES 7.5, 100 mM NaCl buffer for 30 minutes on ice. Conversely, to identify the βarr1-interacting interface on SH3, SH3 (20 uM) was incubated with 5-fold molar excess of βarr1–MC-393 or βarr1–MC-393–V2Rpp in the 20 mM HEPES 7.5, 100 mM NaCl buffer for 30 minutes on ice. The protein stock solution was diluted 17-fold in the 20 mM HEPES 7.5, 100 mM NaCl buffer prepared in D$_2$O. The protein stock solution was diluted 17-fold into D$_2$O-based buffer (20 mM HEPES pH 7.5, 100 mM NaCl) to achieve a final D$_2$O fraction of 94% and to maintain an optimal dynamic range of deuterium uptake, thereby improving the reproducibility of HDX-MS measurements. The deuterium labeling reactions were performed 15 °C. At a designated time point (300, 1000, and 5000 s), the exchange was quenched by adding an equal volume of 0.8% formic acid in H2O, yielding pH 2.6. The sample was digested on

Enzymate BEH Pepsin column (Waters) at a flow rate of 0.15 ml/min using 0.15% formic acid/3% acetonitrile as the mobile phase. An inline 4-µl C8-Opti-lynx II trap cartridge (Optimize Technologies) was used for desalting of the digested peptides. The peptides were then eluted through a C-18 column (Thermo Fisher Scientific, 50 × 1 mm Hypersil Gold C-18) using a rapid gradient from 10 to 90% acetonitrile containing 0.15% formic acid and a flow rate of 0.04 ml/min, leading directly into a maXis-II ETD ESI-QqTOF mass spectrometer (Bruker Daltonics). Up to 30, 150, and 150 pmol of βarr1, SH3, and V2Rpp, respectively, were used in each experiment to analyze βarr1 dynamics and interactions. To assess SH3 dynamics and interactions, up to 30, 150, and 150 pmol of SH3, βarr1, and V2Rpp, respectively, were used. An excess of proteins was included to ensure saturation of complex formation. The total time for the digest and desalting was 3 min, and all peptides had eluted from the C-18 column by 15 min. Pepsin and C-18 column were thoroughly washed after each run. The peptide fragments were identified using Bruker Compass and Biotools software packages. The level of deuterium incorporation was calculated using HDExaminer-3 (Trajan Scientific) from triplicate measurements of each time point. Only regions that exhibited statistically significant differences in deuterium uptake greater than 0.30 Da for free βarr1 and 0.36 Da for βarr1–V2rpp were considered relevant. To improve our confidence in the observed deuterium uptake differences, we sought to increase peptide coverage by analyzing a non-deuterated βarr1 sample. This analysis was performed under the previously employed HDX-LC conditions using a timsTOF fleX MALDI-2 mass spectrometer with ion mobility engaged. The identified peptides were then mapped to the existing HDX dataset. Although peptide overlap in the HDX dataset prevented the analysis of all peptide sequences, this approach ultimately increased the number of analyzable peptides by approximately 30%. Additionally, we conducted further experiments to measure the fully deuterated exchange profiles of βarr1 to estimate back-exchange. To prepare the fully deuterated protein, we first created a 480 µL stock solution of 3 M deuterated GuHCl in $D_2O$. Subsequently, 20 µL of a 20 µM βarr1 protein stock was added to this GuHCl/$D_2O$ solution. The sample was then incubated at 42 °C for three days to allow for deuterium exchange. The GuHCl was deuterated beforehand by dissolving it in $D_2O$ and lyophilizing the solution to dryness. The peptides exhibited back exchange values ranging from 10% to 53%, with an average of 31%.

### M2-FLAG pull-down assay

To test the binding between βarr1 and the SH3 domain of Src, βarr1 (20 µM) was incubated with 3-fold molar excess of V2Rpp for 30 minutes at room temperature, then 10 µM of SH3-FLAG was added. 30 µl of anti-FLAG M2 affinity gel (Millipore Sigma) was added thereafter and the mixture was incubated for 1 hour at room temperature with rotation. After incubation the anti-FLAG M2 resin was collected by centrifugation and washed with 1 ml of 20 mM HEPES pH 7.5, 100 mM NaCl buffer three times. The proteins were eluted with 0.2 mg/ml FLAG-peptide in 20 mM HEPES pH 7.5, 100 mM NaCl buffer, then mixed with Laemmli sample buffer (BioRad), subjected to SDS-PAGE and Western blotting and detected by monoclonal ANTI-FLAG M2-peroxidase (HRP) antibody (1:2000) (A8592, Sigma-Aldrich, RRID: AB_439702) for SH3-FLAG and polyclonal A1CT antibody[58] (1:5000) and HRP-conjugated ECL Rabbit IgG (1:5000) (NA9340, Amersham, RRID:AB_772191) for βarr1.

### In vitro Src C-tail phosphorylation followed by M2-FLAG pull-down assay

Purified Src (20 µM) was incubated with 4 µM Csk in the presence of 100 µM ATP and 5 mM MgCl2 for 2 hours at room temperature. A control reaction was prepared identically, excluding ATP and MgCl2. The C-tail phosphorylation of Src was assessed by Western blotting using Anti-Src (phospho Y529) antibody (ab32078, Abcam, RRID:

AB2286707). The samples of C-tail phosphorylated and unphosphorylated Src (20 µM) were added to 7 µM of βarr1-FLAG pre-incubated with 35 µM of V2Rpp. 30 µl of anti-FLAG M2 affinity gel (Millipore Sigma) was added thereafter and the mixture was incubated for 1 hour at room temperature with rotation. After incubation the anti-FLAG M2 resin was collected by centrifugation and washed with 1 ml of 20 mM HEPES pH 7.5, 100 mM NaCl buffer three times. The proteins were eluted with 0.2 mg/ml FLAG-peptide in 20 mM HEPES pH 7.5, 100 mM NaCl buffer, then mixed with Laemmli sample buffer (BioRad), subjected to SDS-PAGE and Western blotting. The total Src was detected by anti-Src antibody (1:5000) (EMD Millipore 05-184, RRID: AB_2302631) and HRP-conjugated ECL Mouse IgG (1:5000) (NA9310, Amersham, RRID:AB_772193). βarr1-FLAG was detected by monoclonal ANTI-FLAG M2-peroxidase (HRP) antibody (1:2000) (A8592, Sigma-Aldrich, RRID: AB_439702). Western Blot images were taken using BioRad ChemiDoc system and the densitometry analysis was performed by ImageLab v6.1.

### Cross-linking mass-spectrometry

Purified βarr1 (90 µM) was incubated the purified SH1 domain of Src (90 µM) for 1 hour at room temperature. Cross-linking reagents (disuccinimidyl suberate (DSS), Bis(sulfosuccinimidyl) suberate (BS3), dithiobis(succinimidylpropionate) (DSP), 3,3´-dithiobis(sulfosuccinimidylpropionate) (DTSSP), ethylene glycol bis(succinimidyl succinate)) (EGS)) were added thereafter to a final concentration of 2 mM. The reactions were incubated for 30 min and quenched by 50 mM Tris pH 8.0 for 15 min. The proteins were subjected to SDS-PAGE and visualized by Ready Blue Coomassie stain (Sigma-Aldrich). The bands of interest (80-150 kDa) were excised from the gel and subjected to destaining using 25 mM ammonium bicarbonate in 50% acetonitrile. Destained gel slices were dehydrated using 100% acetonitrile and dried using speed-vac for 5 min to remove remaining acetonitrile. 10 mM dithiothreitol and 55 mM iodoacetamide were used for reduction and alkylation of cysteine residues, respectively. The gel slices were washed again in 25 mM ammonium bicarbonate and dried using 100% acetonitrile following by 5-min speed-vac evaporation. The proteins were then subject to in-gel trypsin digestion with 10 ng/µl trypsin (Sequencing grade, Promega) in 25 mM ammonium bicarbonate and overnight at 37 °C. The following day, the tryptic peptides were extracted with 100% acetonitrile and lyophilized. The peptides were loaded onto EvoSep tips (EvoSep) following manufacturer's instructions. The peptide samples were analyzed using a timsTOF Pro 2 mass spectrometer (Bruker) coupled with an Evosep One LC system (Evosep) connected to an 8-cm Evosep performance column (EV-1109) using a pre-defined 30-samples-per-day extended gradient. The following parameters were applied for data-dependent acquisition: intensity threshold: 2.5E3; charge states: +1 to +5; dynamic exclusion: 0.4 min; target intensity for fragmentation: 2E4; isolation window (linear): 2 m/z at 700 m/z and 3 m/z at 800 m/z; collision energy (linear): 20 eV at 1/k0 of 0.60 V·s/cm2 and 59 eV at 1/k0 of 1.60 V·s/cm2. The cross-linked peptides were identified using MaxQuant v.2.6.3.0 with following parameters: digestion mode Trypsin/P; maximum missed cleavages 3; minimum length for a paired-peptide sequence 6; minimum score for cross-linked peptides 100; minimum matches 3; first search peptide tolerance 20 ppm; main search peptide tolerance 10 ppm; isotope match tolerance 0.005 Da, centroid match tolerance 10 ppm; maximum charge 6; isobaric weight exponent 0.75.

### GST pull down assay

To test the binding between different constructs of GST-βarr1 (full-length βarr1, βarr1-N domain (1-176), βarr1-C domain (177-420)) and the SH1 domain of Src, GST-βarr1 (3 µM) was incubated with SH1 (9 µM) for 1 hour at room temperature. 50 µL of GST-beads (GoldBio) equilibrated in 20 mM HEPES pH 7.5, 100 mM NaCl buffer were added thereafter, and the mixture was incubated for 1 hour at room

temperature with rotation. After incubation the GST beads were collected by centrifugation and washed with 20 mM HEPES pH 7.5, 150 mM NaCl buffer three times. The proteins were eluted from GST beads with 100 µL of 50 mM reduced glutathione in 20 mM HEPES pH 8.0, 150 mM NaCl buffer, then mixed with 4x Laemmli sample buffer (BioRad), subjected to SDS-PAGE and Western blotting. SH1 was detected by polyclonal HRP-conjugated Anti-6X His-tag antibody (1:2000) (ab1187, Abcam, RRID: AB_298652); βarr1 was detected by monoclonal HRP-conjugated GST-tag antibody (1:2000) (8-326) (MA4-004-HRP, ThermoFisher, RRID: AB_2537634).

### In vitro Src continuous colorimetric kinase assay

Continuous colorimetric kinase assay[73] was performed in 100 mM HEPES 7.5, 150 mM NaCl, 5 mM MgCl2, 0.005% Triton X-100, containing 0.25 mM optimal Src peptide (AEEEIYGEFEAKKKK), 1 mM phosphoenolpyruvate, 0.3 mM NADH, 4 units of pyruvate kinase and 6 units of lactic dehydrogenase. The concentration of Src was 20 nM, the concentrations of βarr1 and V2Rpp were 100 nM and 200 nM, respectively. Reactions were started by the addition of ATP to a final concentration of 0.1 mM, and the decrease in NADH absorbance was monitored over 40 min at 25 °C using a CLARIOstar microplate reader (BMG Labtech). The initial velocity of the reaction (V0) determined using a linear regression curve fit (GraphPad Prism v9) and was converted to the amount of product formed in the reaction volume per minute using the Beer-Lambert law. Statistical comparisons were determined by one-way ANOVA with Dunnett's post hoc test.

### Src activation assay in HEK-293 βarr1/βarr2 dKO cells

HEK-293 βarr1/βarr2 dKO cells[28] were co-transfected with receptor (FLAG-D1R or chimeric FLAG-β2V2R), Src (only for chimeric FLAG-β2V2R) and wild-type or mutant βarr1 with a C-terminal HA tag with a 1:5 DNA:FuGENE®6 (Promega) ratio according to the manufacturer's instructions. All experiments were conducted 48 hours after transfection. Cells were serum starved for 16 hours in MEM supplemented with 1% penicillin-streptomycin and 0.1% (w/v) bovine serum albumin. The assay was initiated with 5-minute stimulation (10 µM dopamine for D1R) or 10-minute stimulation (10 µM BI-167107 for β2V2R) at 37 °C. The medium was removed, cells were placed on ice and lysed with 2x Laemmli sample buffer (BioRad) supplemented with 4% β-mercaptoethanol. Samples of equal volume from cell lysates were separated with SDS-PAGE, transferred to nitrocellulose membranes and immunoblotted. Primary antibodies and dilution used are as follow: monoclonal ANTI-FLAG M2 peroxidase (HRP) antibody (1:2000) (A8592, Sigma-Aldrich, RRID: AB_439702) to detect β2V2R, polyclonal A1CT antibody generated in Lefkowitz lab[58] (1:5000) for wild-type and mutant βarr1, anti-Src monoclonal antibody (1:1000) (MA5-15214, Thermo Fisher Scientific, RRID: AB_10980540) for total Src, anti-Src polyclonal Y418 for overexpressed Src (1:5000) (ab4816, Abcam, RRID: AB_304652), and Phospho-Src family polyclonal Y416 for endogenous Src (1:10000) (2101, Cell Signaling, RRID: AB_331697). Secondary antibody included HRP-conjugated ECL Rabbit IgG (1:5000) (NA9340, Amersham, RRID:AB_772191) and HRP-conjugated ECL Mouse IgG (1:5000) (NA9310, Amersham, RRID:AB_772193). Western Blot images were taken using BioRad ChemiDoc system and the densitometry analysis was performed by ImageJ v1.52a and ImageLab v6.1.

### Bimane fluorescence

Purified βarr1-MC-393 V70C was labeled overnight at 4 °C with a 3-fold molar excess of monobromobimane (mBr) (Sigma-Aldrich) and an additional 3-fold molar excess added for 1 h at room temperature the next day. Reactions were quenched with L-cysteine, and free mBr was removed by SEC. The experiments were performed using membranes with phosphorylated V2R and purified chimeric M2V2R reconstituted in lipid nanodiscs. Aliquots of phosphorylated V2R membranes were activated with AVP (10µM) then incubated with βarr1 V70C-mBr

(20 nM), and SH3 (250 nM) in the assay buffer (20 mM HEPES 7.4, 100 mM NaCl, 1 mg/ml bovine serum albumin) for 120 min at room temperature in black solid-bottom 96-well microplates (Corning) with gentle agitation. Purified M2V2R reconstituted in lipid nanodiscs (25 nM) was activated with 10 µM iperoxo and subsequently incubated for 60 min at room temperature with 1.5-fold molar access of bimane-labeled βarr1, 2-fold molar excess of Fab30 and SH3 (500 nM). Fluorescence emission spectra were collected in top-read mode, with excitation at 370 nm (16 nm bandpass) and emission scanning from 410 nm to 640 nm (10 nm bandpass) in 0.5 nm increments using a CLARIOstar microplate reader (BMG Labtech). Statistical comparisons were determined by comparing the area under the curves (GraphPad Prism v9) and peak fluorescence using one-way ANOVA with Dunnett's post hoc test from n = 3 (M2V2R) or n = 6 (V2R membranes) independent experiments.

### Fluorescence Resonance Energy Transfer (FRET)-based live-cell cAMP accumulation assay

cAMP production mediated by Gαs-coupled β2AR activation was measured using Exchange Protein Activated by cAMP 2 (Epac2) sensor containing a CFP and YFP FRET pair[31]. HEK293 cells stably expressing Epac2 and β2AR were transfected with Src (2 ug) or the equivalent amount of control plasmids (pcDNA 3.1) with FuGENE®6 (Promega) transfection reagent according to the manufacturer's instructions. 20 hours after transfection the cells were plated in poly-D-lysine-coated, black, clear-bottom 96-well plates (Corning) at a density of 50,000 cells per well. 24 hours after plating, cells were washed with PBS and incubated in imaging buffer (10 mM HEPES, 150 mM NaCl, 5 mM KCl, 1.5 mM MgCl2, 1.5 mM CaCl2, 10 mM glucose, 0.2% BSA, pH 7.4) for one hour at 37 °C. The cells were stimulated with 10 µM isoproterenol (ISO), and FRET changes were measured in real-time using FlexStation 3 plate reader (Molecular Devices). The changes in the background-subtracted 480 nm/535 nm fluorescence emission ratio (CFP/YFP) are indicative of changes in cAMP levels. To quantify the overall cAMP response, the area under the curve (AUC) was calculated from the time-course data in GraphPad Prism v9.

### Statistics and reproducibility

Data were analyzed using GraphPad Prism v9. Data represent the mean ±SEM of at least three independent experiments. Statistical significance for more than two groups was determined by one-way ANOVA with Dunnett's post hoc test. Statistical significance for two groups was determined by unpaired two-tailed Student's t-test or two-tailed Welch's t-test. No samples or data points were excluded from analysis. Uncropped and unprocessed scans of the blots are provided in the Source Data file.

The materials presented are available upon request from Robert J. Lefkowitz (lefko001@receptor-biol.duke.edu).

### Reporting summary

Further information on research design is available in the Nature Portfolio Reporting Summary linked to this article.

## Data availability

The cryo-EM maps have been deposited in the EMDB under accession codes EMD-45977 (SH3–βarr1-CC complex), EMD-45982 (SH3–βarr1-N complex), and EMD-44881 (Src–βarr1-CC complex). The atomic coordinates have been deposited in the Protein Data Bank under accession codes 9CX3 (SH3–βarr1-CC complex); 9CX9 (SH3–βarr1-N complex); 9BT8 (Src–βarr1-CC complex). The HDX-MS data have been deposited to the ProteomeXchange Consortium [http://proteomecentral.proteomexchange.org] via the MassIVE repository [https://massive.ucsd.edu/] with the dataset identifier PXD073493. CXMS data have been deposited to the ProteomeXchange Consortium [http://proteomecentral.proteomexchange.org] via the PRIDE partner

repository[74] with the dataset identifier PXD073058. All other data generated or analyzed in this study are included in the article and its Supplementary Information. Source data are provided within the Source Data File. The manuscript refers to the following previously published PDB accession codes: 4JQI (βarr1–V2Rpp); 2PTK (Src); 1G4M (βarr1); 1FMK (Src); 6TKO (β1V2R–βarr1); 6U1N (M2V2R–βarr1); 6UP7 (NTSR1–βarr1); 4U5W (Hck–Nef); 2KNB (endophilin A1 SH3–parkin Ubl); 3A98 (DOCK2 SH3–ELMO 1); 1JT4 (Sla1 SH3-ubiquitin); 1Y57 (Src). Source data are provided with this paper.

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

## Acknowledgements

We acknowledge the use of cryo-EM microscopes at the Shared Materials Instrumentation Facility (Duke University) and thank Nilakshee Bhattacharya for assistance with microscope operation. We are grateful to Liyin Huang for their helpful discussions throughout this work and to Yangyang Li for administrative assistance. R.J.L. is an Investigator of the Howard Hughes Medical Institute. This work was supported, in part, by US National Institutes of Health (National Heart, Lung, and Blood Institute: R01 HL16037 to R.J.L.; National Institute of General Medical Sciences: R35GM133598 to A.G.). N.P. is supported by postdoctoral fellowships from Human Frontier Science Program (LT000174/2018) and European Molecular Biology Organization (ALTF 1071-2017). K.X. is supported by the Moonshot Biomarker Program of Allegheny Health Network Cancer Institute and Highmark Health, the Prostate Cancer Foundation Challenge Award (2023CHAL4223), the PA State Formula Grant (SAP #: 4100095527), and The Pittsburgh Foundation (cc#45126409).

## Author contributions

N.P. and R.J.L. conceived the study. N.P., R.R.A., J.K., A.W.K., B.P., K.X., and S.A. designed the experiments. N.P., B.N.T., L.L., D.K.B., A.W.K., B.P., K.X., R.O., and X.Z. performed biochemical experiments. N.P., B.N.T., J.K. and S.A. performed cellular functional assays. N.P., B.N.T., and D.K.B. performed structural biology experiments. N.P. and H.B. built and refined structural models. R.R.A. and S.L. performed HDX-MS experiments. N.P., B.N.T., H.B., D.K.B., R.R.A., J.K., A.W.K., B.P., and K.X. analyzed the data. N.P. wrote the manuscript with input from all authors. A.G. and R.J.L. supervised the work.

## Competing interests

The authors declare no competing interests.
