## [Transparent Peer Review file · Nature Communications]

Mechanism of beta-arrestin 1 mediated Src activation via Src SH3 domain revealed by cryo-electron microscopy

Corresponding Author: Professor Robert Lefkowitz

Version 0:

Reviewer comments:

Reviewer #1

(Remarks to the Author)

In this manuscript, Pakharukova et al showed that b-arr1 interacts with the Src SH3 domain via two distinct sites (N- and CC-) and activates the auto-inhibited Src, thereby plays as an active signal transducer. Authors have identified two distinct sites of the Src SH3 domain that bind to b-arr1, determined the complex structures using cryoEM, and performed various biochemical and mutational analysis to support the activation mechanism of Src by b-arr1. They properly addressed concerns raised by previous reviewers and the revised manuscript provides insights into understanding the molecular mechanism by which Src activated by b-arr1.

Although most of the concerns by the previous reviewers have been addressed, I am bit confused on the mutational analysis that performed to understand the physiological meaning of the current study (Fig 7 and lines 275 ~ 288).

Unlike the authors described in the discussion session

"... this configuration enables β arr1 to ensure robust recruitment and activation of Src when one of the interaction sites is sterically hindered or occupied by a competing binding partner...", I noticed that mutation of either site exhibited similar to that of the b-arr1 lacking effect in Fig 7.

Q1. Fig 7a, mutation of both N- and CC sites (mut 3) exhibited higher normalized density for pSrc compared to mut 1 or mut 2. Perhaps, interaction sites are broadly distributed and multiple mutation may not provide decisive effects. Nevertheless, after reading the entire manuscript, the mutation studies are puzzling. Authors may comment on their mutational analysis.

Q2. Mutation on one of the interaction sites should exhibit some degree of activation. In Fig 7c, mutation on N-site (mut 1) or CC-site (mut 5 or 6) exhibited an effect similar to reaction in the absence of b-arrestin. Authors should comment on this.

Related to above Q. Lines 285~290

"Whereas β -arr1 WT (activated by V2Rpp and Fab30) increases Src activity two-fold, β arr1-N mutant P88G_P91G, .. as well as P88G_P91G_P121E_P124G mutant failed to activate Src (Fig. 7c). β arr-CC mutants P121E_P124G and F80A only slightly reduced Src activation compared to β arr1 WT. Therefore, this supports a mechanism where P88 and P91 drive Src activation via the β arr1-N site and F75, N122 and I314 are critical for Src activation via the β arr1-CC site."

Why N-mutant failed to increase the Src activation despite it has functional CC-site?

Q3. Line 281-282

"..In contrast, none of the β arr1 mutants showed a response to dopamine stimulation, suggesting their reduced ability to activate Src (Fig. 7b)..."

The band intensity of each line is difficult to quantify by eyes. But mutant 2 (cc-mutant) showed similar increase to that of the WT (and showed some response). Authors may consider to revise this part for clarity.

Reviewer #2

(Remarks to the Author)

Please find my technical review of the manuscript by Pakharukova et al. below. The current presentation and analysis of the HDX-MS data are insufficient to support the main conclusions of the study. The data as presented do not meet the current standards for reporting and validation in the field, which prevents a full and fair evaluation of the work. The suggestions below focus on areas relevant to HDX-MS experiments that should be clarified and expanded to enhance the paper's impact.

For this reason, I cannot recommend publication in its present form. However, the concerns are addressable. I would be willing to re-evaluate a revised manuscript that thoroughly addresses the major points listed below.

Major points:

The central issue with the manuscript is the lack of a comprehensive and transparent presentation of the HDX-MS data. Without this, the technical quality of the experiment and the statistical validity of the conclusions cannot be verified.

The manuscript currently presents deuterium uptake data for a small subset of peptides (three in Fig. 3, two in Supp. Fig. 5). While these examples are illustrative, this selective presentation makes it difficult to assess the overall quality and robustness of the entire dataset. No other peptides are presented that show no differences between protein states being compared. To build confidence in the findings, the data must be presented more comprehensively:

- To establish the technical quality of the experiment, the manuscript must be expanded to include standard HDX-MS quality metrics. Specifically, please report the following for the final, quality-controlled dataset: The total number of identified peptides, protein sequence coverage (%), average peptide redundancy. These metrics are readily available in HDExaminer and are essential for a complete report.
 - The authors state that significance was defined as a deuterium uptake difference of >0.2 Da across at least two peptides or time points. This is a very small difference. To validate this, please clarify:
 - How many peptides in total from the experiment met this significance criterion?
 - Are the identified regions of differential uptake supported by multiple, overlapping peptides? This is a key part of validating HDX-MS findings.
 - To provide the necessary evidence for the claims made, a comprehensive view of the dataset is required. Two standard approaches would be acceptable:
 - Include a comprehensive data table that displays the D-uptake values for all identified peptides, following established reporting guidelines (e.g., as shown in Masson et al., 2019: <https://pubmed.ncbi.nlm.nih.gov/31249422/>).
 - A heatmap with differential D-uptake between states would prove useful. Alternatively, provide the individual deuterium uptake plots for all peptides identified as part of the Supplementary Information.
- Addressing the points above by providing a complete and transparent view of the dataset is essential to support findings.

Minor points:

- Statistical significance (p-values) is missing from Supplementary Fig. 5. Is this because the differences shown are not statistically significant?
- Please include the amount of protein (in pmol) that was injected on-column for the analysis.
- Please clarify the labeling temperature, as this is not clear. Was the deuterium labeling reaction performed on ice or at another controlled temperature?
- A final but important point on technical accuracy: The manuscript repeatedly describes HDX-MS as a measure of 'solvent accessibility' (e.g., line 142). This phrasing should be revised throughout the text. HDX rates are primarily governed by the stability of backbone hydrogen bonds and local protein dynamics, not simply by solvent exposure. A more accurate description of the goal would be 'to map the interaction interface' or 'to probe changes in protein conformation'. Please ensure this correction is applied consistently for accuracy.

Reviewer #3

(Remarks to the Author)

Version 1:

Reviewer comments:

Reviewer #1

(Remarks to the Author)

The authors have adequately resolved concerns raised by this reviewer.

Reviewer #2

(Remarks to the Author)

Reviewer #3

(Remarks to the Author)

Point-by-point responses to the reviewers

Reviewer #1 (Remarks to the Author):

In this manuscript, Pakharukova et al showed that b-arr1 interacts with the Src SH3 domain via two distinct sites (N- and CC-) and activates the auto-inhibited Src, thereby plays as an active signal transducer. Authors have identified two distinct sites of the Src SH3 domain that bind to b-arr1, determined the complex structures using cryoEM, and performed various biochemical and mutational analysis to support the activation mechanism of Src by b-arr1. They properly addressed concerns raised by previous reviewers and the revised manuscript provides insights into understanding the molecular mechanism by which Src activated by b-arr1.

We thank the reviewer for the thorough review of our manuscript and the valuable suggestions.

Although most of the concerns by the previous reviewers have been addressed, I am bit confused on the mutational analysis that performed to understand the physiological meaning of the current study (Fig 7 and lines 275 ~ 288). Unlike the authors described in the discussion session "... this configuration enables β arr1 to ensure robust recruitment and activation of Src when one of the interaction sites is sterically hindered or occupied by a competing binding partner...", I noticed that mutation of either site exhibited similar to that of the b-arr1 lacking effect in Fig 7.

We thank the reviewer for the thoughtful analysis of the mutational studies. As correctly noted, mutations in either the N-site of β arr1 or selected residues in the CC-site abolished β arr-mediated Src activation in the *in vitro* kinase assay (Fig. 7c), consistent with cooperative rather than redundant roles of the two sites. In HEK293 cells (Fig. 7a,b), particularly downstream of the β 2V2R receptor, some mutants nevertheless retained ability to activate Src despite impairment of one site. These findings suggest that the contribution of each site, and the degree of cooperativity between them, is context-dependent, with additional scaffolding interactions and membrane confinement likely providing partial functional compensation.

To clarify that, we have added the following sentences to the Results section (also in response to Q1 and Q2):

"Upon stimulation of the chimeric β 2-adrenergic receptor containing the V2R tail (β 2V2R) with BI-167107, β arr1 mutants exhibited a moderate reduction in β arr-dependent Src activation (Fig. 7a), indicating partial redundancy between the two Src-binding sites in this context."

"Interestingly, in the *in vitro* assay, mutation of either the β arr1-N site or selected CC-site residues completely abolished Src activation despite the presence of the other intact site. This finding indicates that in the minimal *in vitro* reconstitution system the two sites act cooperatively rather than independently: loss of one site compromises the ability of the other to promote activation. In cells, however, single-site mutants (all mutants downstream of β 2V2R; P121E_P124G downstream of D1R) still retained partial ability to activate Src (Fig. 7a, b), suggesting that the relative contribution of each site and the extent of cooperativity between them are context-

dependent, with additional scaffolding interactions and membrane confinement likely providing partial functional compensation.”

We have also revised the Discussion as following:

“Although Src contains a single SH3 domain and can engage only one β arr1 site at a time, the presence of two spatially proximal SH3-binding motifs within β arr1 is likely to enhance Src recruitment through avidity-like effects. Rather than functioning as redundant, independent docking sites, the two weak-affinity sites increase the effective local concentration of SH3 ligands and promote rapid rebinding following dissociation, thereby stabilizing what would otherwise be transient interactions. This model explains why mutation of either site abolishes Src activation in the minimal *in vitro* system, where cooperative rebinding is essential, while single-site mutants retain partial activity in cellular contexts, where additional scaffolding interactions and membrane confinement can partially compensate. Thus, the dual-site architecture of β arr1 appears optimized to enhance the kinetics and stability of Src engagement rather than to provide simple redundancy.”

Q1. Fig 7a, mutation of both N- and CC sites (mut 3) exhibited higher normalized density for pSrc compared to mut 1 or mut 2. Perhaps, interaction sites are broadly distributed, and multiple mutation may not provide decisive effects. Nevertheless, after reading the entire manuscript, the mutation studies are puzzling. Authors may comment on their mutational analysis.

We thank the reviewer for this comment. We agree with the reviewer’s suggestion that, in the case of mutant 3 (P88G_P91G_P121E_P124G), these mutations did not exert a decisive effect. This is likely because residues F75, N122, and I314 are critical for Src activation via the CC-site (as shown in the *in vitro* experiment). Consequently, mutant 3 still retained some ability to activate Src.

To clarify that, we have revised the Results section as follows (with the added text underlined):

We generated four β arr1 mutants targeting these regions and assessed Src activation in HEK-293 β arr1/ β arr2 double-knockout (CRISPR-Cas9-based, dKO) cells²⁸ by monitoring phosphorylation of the activation loop tyrosine Y416 following GPCR stimulation. Upon stimulation of the chimeric β 2-adrenergic receptor containing the V2R tail (β 2V2R) with BI-167107, β arr1 mutants exhibited a moderate reduction in β arr-dependent Src activation (Fig. 7a), indicating partial redundancy between the two Src-binding sites in this context. Notably, mutant 3, harboring combined mutations in both the β arr1-N and β arr1-CC sites (P88G_P91G_P121E_P124G), did not further decrease Src activation compared with mutant 1 (P88G_P91G) or mutant 2 (P121E_P124G). This finding suggests that β arr1–Src interactions are broadly distributed, and that the mutations tested, while disruptive, are not sufficient to fully abolish β arr-mediated Src activation.

Q2. Mutation on one of the interaction sites should exhibit some degree of activation. In Fig 7c, mutation on N-site (mut 1) or CC-site (mut 5 or 6) exhibited an effect similar to reaction in the absence of b-arrestin. Authors should comment on this.

We thank the reviewer for this suggestion. These results point to cooperativity between the two sites in the minimal *in vitro* system, whereby loss of one site impairs activation mediated by the other.

To address this point, we have revised the Results section as follows (with the added text underlined):

To dissect the contribution of individual residues, we generated six β arr1 mutants and assessed their ability to mediate Src activation *in vitro* by monitoring phosphorylation of a Src peptide substrate, using V2Rpp and Fab30 to activate β arr1. Whereas β arr1 WT increased Src activity approximately two-fold, the β arr1-N mutant (P88G_P91G), β arr1-CC triple/quadruple mutants (P121E_P124G_F75A_N122A and P121E_P124G_F75A_I314A), as well as the double-site mutant (P88G_P91G_P121E_P124G), failed to activate Src (Fig. 7c). In contrast, CC-site mutants P121E_P124G and F80A only modestly reduced Src activation relative to WT. These results support a model in which residues P88 and P91 mediate activation via the β arr1-N site, while F75, N122, and I314 are essential for activation via the β arr1-CC site. Interestingly, in the *in vitro* assay, mutation of either the β arr1-N site or selected CC-site residues completely abolished Src activation despite the presence of the other intact site. This finding indicates that in the minimal *in vitro* reconstitution system the two sites act cooperatively rather than independently: loss of one site compromises the ability of the other to promote activation. In cells, however, single-site mutants (all mutants downstream of β 2V2R; P121E_P124G downstream of D1R) still retained partial ability to activate Src (Fig. 7a, b), suggesting that the relative contribution of each site and the extent of cooperativity between them are context-dependent, with additional scaffolding interactions and membrane confinement likely providing partial functional compensation.

We have also revised the Discussion as follows:

“Rather than functioning as redundant, independent docking sites, the two weak-affinity sites increase the effective local concentration of SH3 ligands and promote rapid rebinding following dissociation, thereby stabilizing what would otherwise be transient interactions. This model explains why mutation of either site abolishes Src activation in the minimal *in vitro* system, where cooperative rebinding is essential, while single-site mutants retain partial activity in cellular contexts, where additional scaffolding interactions and membrane confinement can partially compensate. Thus, the dual-site architecture of β arr1 appears optimized to enhance the kinetics and stability of Src engagement rather than to provide simple redundancy.”

Related to above Q. Lines 285–290: “ Whereas β -arr1 WT (activated by V2Rpp and Fab30) increases Src activity two-fold, β arr1-N mutant P88G_P91G, .. as well as P88G_P91G_P121E_P124G mutant failed to activate Src (Fig. 7c). β arr-CC mutants P121E_P124G and F80A only slightly reduced Src activation compared to β arr1 WT. Therefore, this supports a mechanism where P88 and P91 drive Src activation via the β arr1-N site and F75, N122 and I314 are critical for Src activation via the β arr1-CC site.”

Why N-mutant failed to increase the Src activation despite it has functional CC-site?

We believe this reflects the cooperative nature of the two binding sites, particularly in the minimal *in vitro* reconstitution system, where disruption of one site impairs the ability of the other to drive activation. We have clarified this point in the revised Results section (see answer to the question above).

Q3. Line 281-282 “..In contrast, none of the β arr1 mutants showed a response to dopamine stimulation, suggesting their reduced ability to activate Src (Fig. 7b)...” The band intensity of each line is difficult to quantify by eyes. But mutant 2 (cc-mutant) showed similar increase to that of the WT (and showed some response). Authors may consider to revise this part for clarity.

We thank the reviewer for pointing that out.

We have clarified the Results section as follows:

While basal levels of phospho-Src were detected in all non-stimulated cells, dopamine stimulation produced a significant increase in phospho-Src in cells transfected with β arr1 WT. In contrast, most β arr1 mutants failed to respond to dopamine, indicating a reduced ability to activate Src, except for the β arr1-CC mutant P121E_P124G, which exhibited modest activation following stimulation (Fig. 7b). Taken together, both the β arr1-CC and β arr1-N sites contribute to Src activation in HEK293 cells downstream of two receptors, β 2V2R and D1R.

Reviewer #2 (Remarks to the Author):

Please find my technical review of the manuscript by Pakharukova et al. below. The current presentation and analysis of the CThe suggestions below focus on areas relevant to HDX-MS experiments that should be clarified and expanded to enhance the paper's impact. For this reason, I cannot recommend publication in its present form. However, the concerns are addressable. I would be willing to re-evaluate a revised manuscript that thoroughly addresses the major points listed below.

Major points:

The central issue with the manuscript is the lack of a comprehensive and transparent presentation of the HDX-MS data. Without this, the technical quality of the experiment and the statistical validity of the conclusions cannot be verified.

We thank the reviewer for thorough assessment of the HDX data and for the helpful suggestions that enhance the impact of the manuscript. We have addressed the reviewer's concerns as detailed below. According to reviewer's recommendation, we also provide Supporting Data Files 1–6: a comprehensive view of the dataset with the D-uptake values for all identified peptides and the individual uptake plots for all datasets, following established reporting guidelines (Masson et al., 2019).

The manuscript currently presents deuterium uptake data for a small subset of peptides (three in Fig. 3, two in Supp. Fig. 5). While these examples are illustrative, this selective presentation makes it difficult to assess the overall quality and robustness of the entire

dataset. No other peptides are presented that show no differences between protein states being compared.

We thank the reviewer for this comment. We have now included the examples of the peptides that show no differences between the protein states in Fig. 3c and Supplementary Fig. 5c).

To build confidence in the findings, the data must be presented more comprehensively:

- **To establish the technical quality of the experiment, the manuscript must be expanded to include standard HDX-MS quality metrics. Specifically, please report the following for the final, quality-controlled dataset: The total number of identified peptides, protein sequence coverage (%), average peptide redundancy. These metrics are readily available in HDEaminer and are essential for a complete report.**

We have now included HDX-MS quality metrics and the experiment details in Supplementary Table 2.

- **The authors state that significance was defined as a deuterium uptake difference of >0.2 Da across at least two peptides or time points. This is a very small difference. To validate this, please clarify:**

- **How many peptides in total from the experiment met this significance criterion?**
- **Are the identified regions of differential uptake supported by multiple, overlapping peptides? This is a key part of validating HDX-MS findings.**

We thank the reviewer for these insightful comments. To improve our confidence in the observed deuterium uptake differences, we sought to increase peptide coverage by analyzing a non-deuterated β arr1 sample. This analysis was performed under the previously employed HDX-LC conditions using a timsTOF fleX MALDI-2 mass spectrometer with ion mobility engaged. The newly identified peptides were then mapped to the existing HDX dataset. Although peptide overlap in the HDX dataset prevented the analysis of all new peptide sequences, this approach ultimately increased the number of analyzable peptides by approximately 30%. Additionally, we conducted further experiments to measure the fully deuterated exchange profiles of β arr1 to estimate back-exchange. To prepare the fully deuterated protein, we first created a 480 μ L stock solution of 3 M deuterated GuHCl in D₂O. Subsequently, 20 μ L of a 20 μ M β arr1 protein stock was added to this GuHCl/D₂O solution. The sample was then incubated at 42 °C for three days to allow for deuterium exchange. The GuHCl was deuterated beforehand by dissolving it in D₂O and lyophilizing the solution to dryness. The peptides exhibited back exchange values ranging from 10% to 53%, with an average of 31%.

1) Number of peptides meeting the significance criterion: In the comparison between free β arr1 and β arr1 in complex with SH3, 5% of the analyzed peptides exhibited significantly decreased deuterium uptake in the presence of SH3, as defined by our established significance criterion (>0.2 Da across at least two overlapping peptides or time points). In the comparison between activated β arr1-V2rpp and β arr1-V2rpp in the presence of SH3, 5% of peptides showed decreased uptake and 0.5% (1 peptide) showed increased uptake meeting the same criterion. These peptides, together with their average Δ D values and the cumulative Δ D over time, are listed in

Supplementary Table 3. Although the differences in deuterium uptake for peptides corresponding to the β arr1-CC and β arr1-N sites were relatively small (0.2–0.6 Da), likely reflecting the low affinity of the interactions, the changes were reproducible across multiple time points and observed in several overlapping peptides in both free and V2Rpp-activated β arr1.

2) Validation through overlapping peptides: The regions exhibiting differential deuterium uptake are supported by several overlapping peptides that display consistent patterns of change over several time points (Supplementary Table 3). Importantly, these overlapping peptides were identified in both free inactive β arr1 and V2Rpp-activated β arr1 in the presence of SH3, supporting the observed HDX-MS differences. Together, these results corroborate the disulfide trapping data and cryo-EM structures.

We have modified the section on the HDX experiments as follows (with the added or modified text underlined):

To address this, we mapped the interaction interfaces of free β arr1, V2Rpp-activated β arr1, and SH3 using hydrogen–deuterium exchange mass spectrometry (HDX-MS) (Fig. 3; Supplementary Table 2). HDX-MS provides insights into protein interaction interfaces by measuring deuterium uptake in peptides typically 5–15 residues in length. Thus, this approach enables us to verify whether SH3 binds to two distinct regions of β arr1 *in vitro* without disulfide trapping. Approximately 5% of the analyzed peptides displayed a statistically significant decrease in deuterium uptake in the presence of SH3, defined as a change greater than 0.2 Da observed in at least two overlapping peptides or time points (Supplementary Table 3). Consistent with the structures and the pull-down data, β arr1 showed a significant decrease in HDX rate in both the β arr1-CC and β arr1-N sites in the presence of SH3 (Fig. 3a, b; Supplementary Table 3). Although the differences in deuterium uptake for peptides corresponding to the β arr1-CC and β arr1-N sites were relatively small (0.2–0.6 Da), likely reflecting the low affinity of the interactions, the changes were reproducible across multiple time points and observed in several overlapping peptides in both free and V2Rpp-activated β arr1 (Supplementary Table 3; Fig. 3b). Notably, a decrease in deuterium uptake was also observed for residues 42–47, which do not directly interact with SH3 in the structure but are located near the β arr1–N site, suggesting that SH3 binding may indirectly stabilize this region. Interestingly, reduced deuterium uptake was also detected in the hinge region of β arr1 and its C-domain within the C-terminal region (residues 351–388), as well as in residues 257–300 and 328–344 in V2Rpp-activated β arr1 (Fig. 3a). Since Src does not directly interact with the C-domain of β arr1⁹, the reduced deuterium uptake in this region likely arises from long-range allosteric stabilization propagated through SH3 binding.

- To provide the necessary evidence for the claims made, a comprehensive view of the dataset is required. Two standard approaches would be acceptable:
 - Include a comprehensive data table that displays the D-uptake values for all identified peptides, following established reporting guidelines (e.g., as shown in Masson et al., 2019: <https://pubmed.ncbi.nlm.nih.gov/31249422/>).
 - A heatmap with differential D-uptake between states would prove useful. Alternatively, provide the individual deuterium uptake plots for all peptides identified as part of the Supplementary Information.

Addressing the points above by providing a complete and transparent view of the dataset is essential to support findings.

We thank the reviewer for this suggestion.

We now include additional Supporting Data Files to provide the comprehensive view of the dataset:

- 1) Tables with the D-uptake values for all identified peptides for each dataset (Supporting Data Files 1–4), following established reporting guidelines as in Masson et al., 2019.
- 2) The individual deuterium uptake plots for all peptides for each dataset (Supporting Data Files 5-6).

These files are provided for both β arr1 and SH3 datasets.

Minor points:

- Statistical significance (p-values) is missing from Supplementary Fig. 5. Is this because the differences shown are not statistically significant?

We thank the reviewer for this insightful comment. Statistical significance (p-values) was not reported for Supplementary Fig. 5 because only a single HDX-MS experiment was performed for the SH3 states; therefore, statistical testing was not applicable. Nevertheless, the observed differences are reproducible across multiple overlapping peptides and reach magnitudes of up to 1.3 Da at individual time points, supporting the biological relevance of these changes (Supplementary Table 4). Accordingly, we present these data as qualitative yet informative observations for readers. To enhance transparency, comprehensive experimental details have been included in Supplementary Table 2, and the legend for Supplementary Fig. 5 has been updated to clarify that the data are based on a single experiment. Furthermore, Supplementary Fig. 5 has been revised to include peptides showing no consistent changes in deuterium uptake.

We added the following text to the manuscript:

We next performed a qualitative HDX-MS experiment to assess changes in the deuterium uptake of SH3 in the presence of either free or V2Rpp-activated β arr1 (Supplementary Tables 2 and 4; Supplementary Fig. 5). Consistent with the structural data, peptides encompassing the nSrc loop and the 3_{10} helix of SH3 exhibited decreased deuterium uptake, with more pronounced differences observed in the presence of V2Rpp-activated β arr1 (Supplementary Table 4; Supplementary Fig. 5a, b). In contrast, peptides corresponding to the RT loop predominantly showed increased deuterium uptake, likely reflecting local destabilization and enhanced conformational flexibility of the RT loop upon β arr1 binding. Taken together, the HDX-MS data corroborate the disulfide trapping and cryo-EM results, confirming the presence of two SH3-binding sites on β arr1 and validating the SH3– β arr1-CC and SH3– β arr1-N complexes as *bona fide* interaction structures.

- Please include the amount of protein (in pmol) that was injected on-column for the analysis.

Up to 30, 150, and 150 pmol of β arr1, SH3, and V2Rpp, respectively, were used in each experiment to analyze β arr1 dynamics and interactions. To assess SH3 dynamics and interactions, up to 30,

150, and 150 pmol of SH3, β arr1, and V2Rpp, respectively, were used. An excess of proteins was included to ensure saturation of complex formation. This information is now included in Methods section.

- Please clarify the labeling temperature, as this is not clear. Was the deuterium labeling reaction performed on ice or at another controlled temperature?

The deuterium labeling reaction was performed at 15°C (now included in the Methods section and Supplementary Table 2).

- A final but important point on technical accuracy: The manuscript repeatedly describes HDX-MS as a measure of 'solvent accessibility' (e.g., line 142). This phrasing should be revised throughout the text. HDX rates are primarily governed by the stability of backbone hydrogen bonds and local protein dynamics, not simply by solvent exposure. A more accurate description of the goal would be 'to map the interaction interface' or 'to probe changes in protein conformation'. Please ensure this correction is applied consistently for accuracy.

We thank the reviewer for this important point. We have now corrected the phrasing throughout the manuscript.

Reviewer #3 (Remarks to the Author):

We thank the reviewer for the assessment of the manuscript and valuable suggestions.

Point-by-point responses to the reviewers

Reviewer #2 (Remarks to the Author):

I thank the authors for their detailed rebuttal and for providing the comprehensive datasets, including the D-uptake values and peptide plots. The inclusion of these data significantly improves the transparency of the study.

After reviewing the full dataset and experimental parameters, the authors need to further refine the interpretation of the D-uptake differences. Specifically, the experimental conditions (high dilution) appear to have limited the target occupancy, resulting in small shifts that are close to the noise (see comments below). While the HDX-MS data provides valuable validation, the current interpretation overstates the statistical certainty of the results given these experimental limitations.

I do not object to the publication of this work; however, I recommend adjusting the text to frame these results as "consistent qualitative trends" that corroborate the CryoEM model, rather than a quantitative definition of the interface. This shift in presentation, along with the data reporting clarifications detailed below, will strengthen the manuscript.

We thank the Reviewer for the careful evaluation of the HDX datasets. In response to this suggestion, we have adjusted the text in the manuscript to present the HDX results as consistent qualitative trends that corroborate the cryo-EM structures:

"Taken together, the HDX-MS data reveal consistent qualitative trends that corroborate the disulfide trapping and cryo-EM results and support the presence of two SH3-binding regions on β arr1 and the interaction modes captured in the SH3- β arr1-CC and SH3- β arr1-N complexes."

Major technical concerns

□ **Target occupancy during HDX:** The authors state in the Methods that SH3 was added in 5-fold excess to ensure saturation. However, the dilution step into D₂O significantly alters the equilibrium. Based on the stated K_d of 6 μ M and the stock concentrations of 20 μ M and 100 μ M, the initial occupancy during equilibration is high (~93%). However, the protocol describes a 17-fold dilution with D₂O. Under these final conditions, the calculated target occupancy of β arr1 (when saturated with SH3) or SH3 (when saturated with β arr1) drops to ~46%. This low occupancy likely explains why the observed differences are so small and why many peptides show no change. A smaller dilution (e.g., 5-fold) would maintain higher occupancy (~70%). Because the occupancy is low, the authors must acknowledge this limitation in their text. The small magnitude of changes is likely an artifact of the experimental design rather than interaction dynamics.

We thank the Reviewer for highlighting this important point. We agree that, for such weak interactions, it is desirable to maximize both complex occupancy and the D₂O fraction during exchange, while minimizing back-exchange in solution and in the gas phase. In this study, we prioritized maintaining 94% D₂O fraction at 17-fold dilution to preserve optimal dynamic range

and improve the reproducibility of HDX measurements. Although this results in a lower complex occupancy (~46%), the improved fidelity of deuterium uptake data provides greater confidence in identifying bona fide interaction sites, particularly given the small Δ HDX values expected for weak complexes. We therefore assumed that the potential gain in occupancy at a 5-fold dilution would be outweighed by the reduced HDX data quality associated with lowering the D₂O fraction to ~80%.

To clarify this for readers, we have included the following statement in the Methods sections (added text shaded in yellow):

“The protein stock solution was diluted 17-fold into D₂O-based buffer (20 mM HEPES pH 7.5, 100 mM NaCl) to achieve a final D₂O fraction of 94% and to maintain an optimal dynamic range of deuterium uptake, thereby improving the reproducibility of HDX-MS measurements.”

And in the Results section:

“Although the differences in deuterium uptake for peptides corresponding to the β arr1-CC and β arr1-N sites were modest (0.2–0.8 Da), consistent with low-affinity interactions and partial complex occupancy arising from dilution during HDX labeling, the changes were consistent across multiple time points and supported by several overlapping peptides in both free and V2Rpp-activated β arr1 (Supplementary Table 3; Fig. 3a, b; Supplementary Fig. 5a).”

□ **Overlapping peptides:** There are inconsistencies in the data where longer peptides show "significant" differences (highlighted in yellow in table below), but shorter, internal peptides covering the same region do not. For example, Supplementary Table 3 highlights the peptide 76–109 as significantly different. However, multiple shorter peptides nested within this sequence (covering the interacting region FRKDLF, residues 80–88) show no difference. Theoretically, the shorter peptides (which are not "diluted" by non-interacting residues) should show larger % D-uptake differences than the longer parent peptide. The fact that they do not, suggests the difference in the longer peptide may be a technical artifact (e.g., integration boundary differences) rather than a biological signal. The authors should re-examine these inconsistencies. If the "significant" hit is not supported by internal overlapping peptides, it should be treated with extreme caution.

We thank the Reviewer for this careful analysis. The apparent discrepancy in deuterium uptake differences between longer and shorter peptides can be explained by several technical and structural factors.

First, the first two N-terminal residues do not contribute to measurable deuterium uptake. Structural analysis indicates that interaction of β arr1 with SH3 in the CC-site relies on residues F75 (hydrophobic interactions) and R76 (two hydrogen bonds). These residues are located at or near the peptide N-terminus in several short internal peptides, such as TFRKDLFVA, FRKDL, FRKDLF, and RKDLF, inherently limiting detectable changes.

Second, the longer peptide RKDLFVANVQSFPPAPEDK spans residues that contribute to both the β arr1-CC site (RKDLF) and the -N site (FPPAP), effectively integrating interactions from two

binding regions. As a result, deuterium uptake differences are more readily detected in this peptide than in shorter fragments that sample only a subset of the interacting surface.

Third, chromatographic behavior further contributes to these differences. Most short peptides (e.g. GLTFRKDLF, FRKDLF, DVLGLTFRKDL, DVLGLTFRKDLF) elute substantially later (~9–13 min) than the longer peptides RKDLFVANVQSFPPAPEDK and LTFRKDLFVANVQ exhibiting significant reduction in HDX uptake (~3 min). Therefore, these shorter peptides are more susceptible to back exchange which further attenuates small differences in deuterium uptake.

To address this issue systematically, we reanalyzed a curated set of peptides encompassing the β arr1-N and -CC interaction sites (residues 66–126) and performed a focused volcano plot analysis (Supplementary Fig. 5a; see below). The significance threshold for this group is 0.2112 Da. This analysis reveals a consistent trend toward decreased deuterium uptake in the presence of SH3 across most peptides, with smaller differences for shorter peptides due to the technical considerations described above. We have now included this analysis as Supplementary Figure 5a and added the following sentence to the Results section:

“Within these regions, longer peptides spanning both β arr1-CC and -N sites showed more pronounced changes, while shorter internal peptides displayed attenuated effects due to key interacting residue being positioned at the peptide N-terminus and increased back exchange (Supplementary Figure 5a).”

Taken together, these observations indicate that the differences observed in the longer peptides reflect peptide length, residue positioning, and chromatographic behavior rather than technical artifacts. Importantly, as suggested by the Reviewer, the HDX data are now presented as qualitative, supportive trends that are fully consistent with the cryo-EM structures:

“Taken together, the HDX-MS data reveal consistent qualitative trends that corroborate the disulfide trapping and cryo-EM results and support the presence of two SH3-binding regions on β arr1 and the interaction modes captured in the SH3- β arr1-CC and SH3- β arr1-N complexes.”

□ **The threshold of 0.2 Da is set at the absolute limit of detection (approximately 2x experimental noise), which increases the risk of false positives. HDExaminer offers multiple representations to evaluate a dataset, such as Residual plot or Wood’s plot, where statistical significance can be automatically calculated based on replicate variance (especially when 3 replicates were performed). What is the automatic cutoff threshold on the Residual plot or Wood’s plot for the β arr1 vs β arr1:SH3 dataset?**

We thank the Reviewer for this helpful suggestion. For the curated set of peptides encompassing the β arr1-N and β arr1-CC interaction sites (residues 66–126), the significance threshold in the volcano plot is 0.21 Da, while the corresponding residual plot cutoff is 0.63 Da (calculated as 0.21 Da multiplied by three data points). For the remainder of the dataset, slightly higher automatic thresholds were obtained, with cutoffs of 0.30 Da for datasets with free β arr1 and 0.36 Da for datasets with V2Rpp-activated β arr1. Given the Reviewer’s concern, we applied a more stringent cutoff across the dataset and report only statistically significant differences exceeding 0.30 Da for β arr1 and 0.36 Da for β arr1–V2rpp at a single time point. Figures 3a and Supplementary Tables 3 and 4 have been revised accordingly.

The following sentences were revised in the Results section:

“Approximately 5% of the analyzed peptides displayed a statistically significant decrease in deuterium uptake in the presence of SH3, defined as a statistically significant change greater than 0.3 Da for free β arr1 and 0.36 Da for β arr1–V2rpp (Supplementary Table 2–3, Supplementary Data 1–2, 5)”.

and in the Methods section:

“Only regions that exhibited statistically significant differences in deuterium uptake greater than 0.30 Da for free β arr1 and 0.36 Da for β arr1–V2rpp were considered relevant”.

□ **Furthermore, given these small magnitudes and potential variance differences between states, a standard Student’s t-test is prone to false positives. The field standard is the Welch’s t-test (Hageman & Weis, 2019, Anal. Chem.), which accounts for unequal variances and is available in HDExaminer. Please verify if the peptides in Figure 3 (residues 76-94, 86-101, 106-126) remain statistically significant using a Welch’s t-test (e.g. they should show as significant in the volcano plot or represented with a star in the D-uptake plot when compared to the control state). If these peptides do not pass the Welch’s test, or given the inconsistencies noted in Point 2, I strongly recommend presenting the HDX-MS data as qualitative trends. This allows the data to successfully serve its purpose (validation of CryoEM) without overstating the quantitative evidence.**

We re-evaluated the statistical significance of peptides in Figure 3b using Welch’s t-test and found that the changes in the presented peptides are statistically significant; they also show as significant in the volcano plot (see above). The exact significance values are now shown in Figure 3b.

Importantly, as suggested by the Reviewer, we frame the HDX data as consistent qualitative trends supporting the cryo-EM structures:

“Taken together, the HDX-MS data reveal consistent qualitative trends that corroborate the disulfide trapping and cryo-EM results and support the presence of two SH3-binding regions on β arr1 and the interaction modes captured in the SH3- β arr1-CC and SH3- β arr1-N complexes.”

Minor points and data presentation

The current tables are incomplete. The authors should re-export all data using the “Export Uptake Summary Table” option in HDExaminer. To ensure the data is reviewable, the table must explicitly contain the following fields: Protein State, Protein, Start, End, Sequence, Peptide Mass, RT, Deut Time, maxD, #D, %D, Conf Interval (#D), #Rep, Confidence, Stddev, and p-value.

We thank the Reviewer for this suggestion. We have now re-exported all data from HDExaminer to include all fields used in the analysis.

It is known that different charge states can sometimes result in different D-uptake values for the same peptide. It appears that for many peptides in this dataset, multiple charge states have been detected (based on the exported plots). The authors must ensure that for direct comparisons between protein states, only similar charge states are taken into account to avoid technical bias.

We thank the Reviewer for the careful analysis of the data. Although multiple charge states were detected, only identical charge states were used for direct comparisons between protein states.

The data exports currently include short peptides that, due to sequence similarity, cannot be distinguished based on MS/MS data. While HDExaminer warns of these peptides with similar masses, it does not automatically exclude them. The authors must manually curate the dataset to identify such cases and include only one peptide representation. A couple of examples found in this dataset include:

β arr1: Residues 42-47 (VDPVG) is indistinguishable from 43-48 (DPVGV)

SH3: Residues 27-30 (VTTF) is indistinguishable from 28-31 (TTFV)

We thank the Reviewer for this comment. We have curated the datasets and included only one peptide representation with the same mass.

The D-uptake plots provided in the supplementary files display Standard Deviation (SD) but lack confidence intervals. Please re-export these plots to include confidence intervals to allow for a proper visual assessment of significance.

We now present the D-uptake plots with confidence intervals.

I strongly encourage the authors to upload their RAW HDX-MS (including identification runs) data to a public repository (e.g., PRIDE or MassIVE). Access to the RAW data is

critical for reviewers to evaluate the manuscript efficiently and fairly and allow for future mining.

We thank the Reviewer for this suggestion. The HDX-MS data have now been deposited to the ProteomeXchange Consortium via the MassIVE repository with the dataset identifier PXD073493 (MSV000100571).

I thank the authors for their detailed rebuttal and for providing the comprehensive datasets, including the D-uptake values and peptide plots. The inclusion of these data significantly improves the transparency of the study.

After reviewing the full dataset and experimental parameters, the authors need to further refine the interpretation of the D-uptake differences. Specifically, the experimental conditions (high dilution) appear to have limited the target occupancy, resulting in small shifts that are close to the noise (see comments below). While the HDX-MS data provides valuable validation, the current interpretation overstates the statistical certainty of the results given these experimental limitations.

I do not object to the publication of this work; however, I recommend adjusting the text to frame these results as "consistent qualitative trends" that corroborate the CryoEM model, rather than a quantitative definition of the interface. This shift in presentation, along with the data reporting clarifications detailed below, will strengthen the manuscript.

Major technical concerns

- Target occupancy during HDX: The authors state in the Methods that SH3 was added in 5-fold excess to ensure saturation. However, the dilution step into D2O significantly alters the equilibrium. Based on the stated K_d of 6 μM and the stock concentrations of 20 μM and 100 μM , the initial occupancy during equilibration is high (~93%). However, the protocol describes a 17-fold dilution with D2O. Under these final conditions, the calculated target occupancy of β arr1 (when saturated with SH3) or SH3 (when saturated with β arr1) drops to ~46%. This low occupancy likely explains why the observed differences are so small and why many peptides show no change. A smaller dilution (e.g., 5-fold) would maintain higher occupancy (~70%). Because the occupancy is low, the authors must acknowledge this limitation in their text. The small magnitude of changes is likely an artifact of the experimental design rather than interaction dynamics.
- Overlapping peptides: There are inconsistencies in the data where longer peptides show "significant" differences (highlighted in yellow in table below), but shorter, internal peptides covering the same region do not. For example, Supplementary Table 3 highlights the peptide 76–109 as significantly different. However, multiple shorter peptides nested within this sequence (covering the interacting region FRKDLF, residues 80–88) show no difference. Theoretically, the shorter peptides (which are not "diluted" by non-interacting residues) should show larger % D-uptake differences than the longer parent peptide. The fact that they do not, suggests the difference in the longer peptide may be a technical artifact (e.g., integration boundary differences) rather than a biological signal. The authors should re-examine these inconsistencies. If the "significant" hit is not supported by internal overlapping peptides, it should be treated with extreme caution.

74	89	EDLDVLGLTFRKDLFV
76	109	LDVLGLTFRKDLFVANVQSFPPAPEDKKPLTRLQ
77	88	DVLGLTFRKDLF
79	88	LGLTFRKDLF
80	88	GLTFRKDLF
81	88	LTFRKDLF
81	90	LTFRKDLFVA
81	93	LTFRKDLFVANVQ
82	90	TFRKDLFVA
83	87	FRKDL
83	88	FRKDLF
84	88	RKDLF
84	102	RKDLFVANVQSFPPAPEDK

- The threshold of 0.2 Da is set at the absolute limit of detection (approximately 2x experimental noise), which increases the risk of false positives. HDExaminer offers multiple representations to evaluate a dataset, such as Residual plot or Wood's plot, where statistical significance can be automatically calculated based on replicate variance (especially when 3 replicates were performed). What is the automatic cutoff threshold on the Residual plot or Wood's plot for the β arr1 vs β arr1:SH3 dataset?
- Furthermore, given these small magnitudes and potential variance differences between states, a standard Student's t-test is prone to false positives. The field standard is the Welch's t-test (Hageman & Weis, 2019, Anal. Chem.), which accounts for unequal variances and is available in HDExaminer. Please verify if the peptides in Figure 3 (residues 76-94, 86-101, 106-126) remain statistically significant using a Welch's t-test (e.g. they should show as significant in the volcano plot or represented with a star in the D-uptake plot when compared to the control state). If these peptides do not pass the Welch's test, or given the inconsistencies noted in Point 2, I strongly recommend presenting the HDX-MS data as qualitative trends. This allows the data to successfully serve its purpose (validation of CryoEM) without overstating the quantitative evidence.

Minor points and data presentation

- The current tables are incomplete. The authors should re-export all data using the "Export Uptake Summary Table" option in HDExaminer. To ensure the data is reviewable, the table must explicitly contain the following fields: Protein State, Protein, Start, End, Sequence, Peptide Mass, RT, Deut Time, maxD, #D, %D, Conf Interval (#D), #Rep, Confidence, Stddev, and p-value.
- It is known that different charge states can sometimes result in different D-uptake values for the same peptide. It appears that for many peptides in this dataset, multiple charge states have been detected (based on the exported plots). The authors must ensure that for direct comparisons between protein states, only similar charge states are taken into account to avoid technical bias.
- The data exports currently include short peptides that, due to sequence similarity, cannot be distinguished based on MS/MS data. While HDExaminer warns of these peptides

with similar masses, it does not automatically exclude them. The authors must manually curate the dataset to identify such cases and include only one peptide representation. A couple of examples found in this dataset include:

βarr1: Residues 42-47 (VDPVG) is indistinguishable from 43-48 (DPVGV)

SH3: Residues 27-30 (VTTF) is indistinguishable from 28-31 (TTFV)

- The D-uptake plots provided in the supplementary files display Standard Deviation (SD) but lack confidence intervals. Please re-export these plots to include confidence intervals to allow for a proper visual assessment of significance.

- I strongly encourage the authors to upload their RAW HDX-MS (including identification runs) data to a public repository (e.g., PRIDE or MassIVE). Access to the RAW data is critical for reviewers to evaluate the manuscript efficiently and fairly and allow for future mining.